# Nucleosome flipping drives kinetic proofreading and processivity by SWR1

Paul Girvan[1,2,4], Adam S. B. Jalal[1,4], Elizabeth A. McCormack[1], Michael T. Skehan[1], Carol L. Knight[1], Dale B. Wigley[1 ✉] & David S. Rueda[2,3 ✉]

The yeast SWR1 complex catalyses the exchange of histone H2A–H2B dimers in nucleosomes, with Htz1–H2B dimers[1–3]. Here we used single-molecule analysis to demonstrate two-step double exchange of the two H2A–H2B dimers in a canonical yeast nucleosome with Htz1–H2B dimers, and showed that double exchange can be processive without release of the nucleosome from the SWR1 complex. Further analysis showed that bound nucleosomes flip between two states, with each presenting a different face, and hence histone dimer, to SWR1. The bound dwell time is longer when an H2A–H2B dimer is presented for exchange than when presented with an Htz1–H2B dimer. A hexasome intermediate in the reaction is bound to the SWR1 complex in a single orientation with the 'empty' site presented for dimer insertion. Cryo-electron microscopy analysis revealed different populations of complexes showing nucleosomes caught 'flipping' between different conformations without release, each placing a different dimer into position for exchange, with the Swc2 subunit having a key role in this process. Together, the data reveal a processive mechanism for double dimer exchange that explains how SWR1 can 'proofread' the dimer identities within nucleosomes.

A canonical nucleosome contains two copies each of the four histones H2A, H2B, H3 and H4 around which approximately 147 bp of DNA are wrapped[4]. However, additional variants have been discovered for each of these histones and, when present, that have special roles in cells, such as at centromeres, and in processes including transcription and DNA repair[5]. Most of these variants are laid down into chromatin during replication, but an exception in yeast is the H2A histone variant Htz1 (H2A.Z in higher eukaryotes). Htz1 is specifically incorporated into nucleosomes by the SWR1 complex[1–3]. In humans, there are two large multi-subunit complexes that incorporate H2A.Z into nucleosomes, SRCAP[6] and TIP60 (ref. 7), the latter complex also being able to acetylate histones, as well as other proteins, as a part of DNA damage signalling[7]. In addition to being signals of DNA damage, nucleosomes that contain Htz1 (or H2A.Z) also have a role in transcriptional regulation[8].

SWR1 is a 14-subunit complex that is a member of the INO80 remodeller family[9]. Cryo-electron microscopy (cryo-EM) structures of INO80 and SWR1 complexes bound to nucleosomes have been reported[10–12]. Despite significant similarity between the complexes in terms of subunits and sequence homology, the two complexes engage with nucleosomes in a very different manner[13]. The ATPase domains of the INO80 subunit engage at superhelical location 6 (SHL6), whereas those of SWR1 are located at SHL2. Both complexes unwrap significant sections of DNA from the nucleosome, but this is stabilized by the motor domains of INO80 (refs. 10,11) and the Arp6–Swc6 subunits in SWR1 (ref. 12). These differences may relate to the differing activities of the two complexes because SWR1, unlike INO80, lacks the ability to slide nucleosomes[14], although

ATP-dependent DNA translocation within the context of the nucleosome wrap is required for activity[14]. Instead, SWR1 catalyses the ATP-dependent exchange of H2A–H2B dimers with those comprising Htz1–H2B[1–3]. The exchange takes place in a stepwise manner with both dimers being exchanged[15]. For canonical nucleosomes, SWR1 shows specificity for exchange of H2A–H2B dimers with Htz1–H2B and will not catalyse the reverse exchange under any conditions so far identified[15,16]. How this remarkable specificity is achieved is unknown, although sequence differences between the α2 helix of H2A and Htz1 probably contribute to this[14]. Acetylation of K56 on H3 in nucleosomes appears to reduce the specificity of histone exchange by interfering with Swc2 function[17].

Single-molecule studies have begun to reveal aspects of the complex process of histone exchange. In the initial complex, when nucleosomes first bind to SWR1, the DNA wrap becomes dynamic with small, but rapid, unwrapping events in addition to the significant unwrapping by Arp6–Swc6 subunits[12]. However, to progress towards dimer exchange, a larger unwrapping occurs[18–20], presumably to fully expose the dimer, although the nature and full extent of this unwrapping remain unclear, as well as which subunits contribute to this process. It is also unclear whether dimer exchange is a processive process, with both dimers exchanged in a nucleosome after a single SWR1-binding event, or is distributive with nucleosome release between dimer exchanges[12,18,20]. If histone exchange is processive, this would suggest a higher propensity for double-exchanged dimer nucleosomes than for single exchanges, although the significance of double-exchanged versus single-exchanged nucleosomes is also unknown.

[1]Section of Structural Biology, Department of Infectious Disease, Faculty of Medicine, Imperial College London, London, UK. [2]Single Molecule Biophysics Group, MRC Laboratory of Medical Sciences, London, UK. [3]Section of Virology, Department of Infectious Disease, Faculty of Medicine, Imperial College London, London, UK. [4]These authors contributed equally: Paul Girvan, Adam S. B. Jalal. ✉e-mail: d.wigley@imperial.ac.uk; david.rueda@imperial.ac.uk

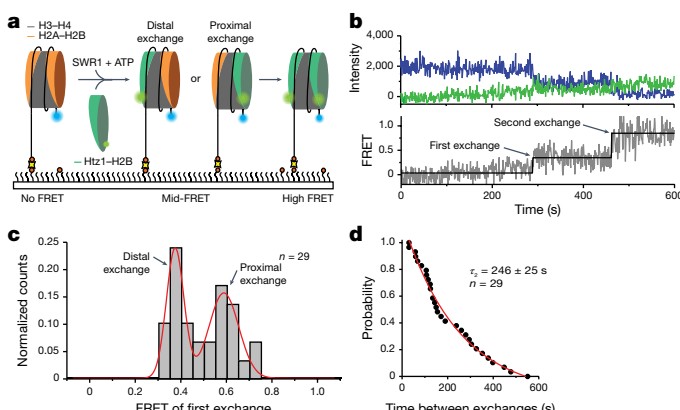

**Fig. 1 | Double-exchange events can be observed by smFRET. a**, Schematic of the assay. Nucleosomes (113N2.AF488) labelled with AF488 (blue) on the short 2-bp overhang are surface immobilized on a PEGylated microscope slide. SWR1, ATP and AF555–Htz1–H2B dimers (green) are flowed in to start the exchange reaction. Histone exchange is detected as a FRET increase between AF488 and AF555. **b**, Intensity trajectory (top) and corresponding FRET trajectory (bottom) for a single nucleosome showing a stepwise gain in FRET signal following each dimer exchange. **c**, Idealized FRET histogram of the first-exchange event shows two approximately equal populations of approximately 0.4 and approximately 0.6 FRET corresponding to either dye-distal or dye-proximal exchange. **d**, Dwell time distribution between the first and second exchanges yields a second-exchange time $\tau_2 = 246 \pm 25$ s. Reported errors are the error of the fit.

## Single-molecule histone exchange by SWR1

We have previously developed a fluorescence resonance energy transfer (FRET)-based assay to monitor histone exchange by SWR1 in bulk phase by monitoring loss of FRET when labelled H2A–H2B dimers are exchanged for unlabelled Htz1–H2B dimers[16]. We have adapted this methodology for single-molecule analysis by changing the dimer labelling so that a gain of FRET was observed when an unlabelled H2A–H2B dimer is exchanged for a labelled Htz1–H2B dimer (Fig. 1a). This makes interpretation of the data less ambiguous, because in a loss of FRET assay, it can be hard to distinguish between histone exchange and dye photobleaching[19]. This assay allows us to monitor two histone dimer exchanges on surface-immobilized nucleosomes as consecutive step increases in FRET (Fig. 1b and Extended Data Fig. 1). A histogram of the average FRET value between the first and second exchange reveals two possible intermediate states at approximately 0.6 and approximately 0.4 FRET (Fig. 1c), which corresponds to donor-proximal and donor-distal exchange, respectively. Control experiments in the presence of non-hydrolysable ATPγS analogue revealed no exchange (Extended Data Fig. 1). A dwell time analysis of the intermediate state (Fig. 1d) shows that the time between the first and the second exchange is $\tau_2 = 246 \pm 25$ s, which is independent of the initial exchange (proximal or distal) and consistent with slow histone exchange observed in other single-molecule studies[18,19] and in ensemble-averaged measurements[2,15,16].

## Double exchange can be processive

The lifetime of SWR1–nucleosome complexes has been shown to be long (several tens of minutes[18]), although is reduced in the presence of ATP[18,19]. We have also determined lifetimes of SWR1–nucleosome complexes to be on the order of tens of minutes (Extended Data Fig. 2). Such long lifetimes, longer than that required for a single histone exchange (2–3 min (refs. 18,19)), raise the possibility of a processive mechanism for double histone exchange, as hinted at previously[19]. However, structural studies[12] have strongly suggested that dimer exchange takes place in the position facing the enzyme complex,

which has the DNA wrap partially unwound by the Arp6–Swc6 subunits. For double dimer exchange to be processive, different mechanisms for exchange would need to occur for each dimer, or a mechanism must exist to rotate the bound nucleosome in situ. An alternative, and seemingly more plausible, possibility is a distributive mechanism that allows the singly exchanged nucleosome to dissociate and then rebind to SWR1 with the appropriate face oriented for dimer exchange. However, it is important to note that any processive enzyme reactions that are observed need to be explained by a different mechanism.

To evaluate these alternatives directly, we expanded our single-molecule FRET (smFRET) assay to three colours with an additional dye (Atto647N) on SWR1 (see Methods) to colocalize enzyme binding and dissociation dynamics (Fig. 2a). The labelled SWR1 did not affect enzyme activity in bulk (Extended Data Fig. 2). Using alternating laser excitation, we can selectively follow histone exchange as stepwise FRET increases, while monitoring SWR1 binding by fluorescence intensity. The resulting single-molecule trajectories showed molecules that undergo single (43%; Fig. 2b) and double (57%; Fig. 2d) exchanges during a SWR1-binding event. In these trajectories, SWR1 binding precedes the first histone exchange by $36 \pm 2$ s (Fig. 2c), whereas the second exchange is approximately six-fold slower, taking $227 \pm 11$ s (Fig. 2e), consistent with the value measured in Fig. 1d. Ultimately, SWR1 dissociates or photobleaches (Fig. 2b,d).

These data demonstrate that SWR1 can exchange two histone dimers in a single-binding event, strongly supporting a processive exchange mechanism. A small fraction (approximately 10%) of distributive exchanges is observed, but this is expected for processive enzymes, as all processive enzymes are expected to exhibit a fraction of distributive events depending on experimental conditions. In some trajectories, we cannot observe the presence of SWR1, probably due to Atto647N photobleaching or incomplete labelling (Extended Data Fig. 3).

## Nucleosomes flip between two bound states

Having recapitulated and monitored the complete double-exchange reaction at the single-molecule level, and established that this can be processive, we then sought to delve more deeply into the different steps of the reaction pathway. We set out to answer two questions: first, how SWR1 determines which dimer to exchange so that H2A–H2B dimers are always replaced with Htz1–H2B dimers and never the reverse; and second, how consecutive exchange reactions are carried out processively without release of the nucleosome.

To answer the first question, we labelled the nucleosome (on the short DNA overhang) with a FRET donor and SWR1 complex with a FRET acceptor (Fig. 3a) to monitor nucleosome dynamics when bound to the complex. The resulting single-molecule FRET trajectories revealed two conformations for bound nucleosomes with different FRET efficiencies (approximately 0.1 and approximately 0.4; Fig. 3b and Extended Data Fig. 4a–d). The cryo-EM structure of the SWR1–nucleosome complex[12] was used to evaluate the nature of the complexes, and the simplest interpretation is that binding of the nucleosome is in two pseudo-symmetric conformations, with each conformation presenting a different dimer to the surface of the SWR1 complex (Fig. 3a). These two bound states place the two dyes either close or further apart, termed dye-proximal and dye-distal conformations, respectively. Most molecules (68%; $n = 154$) showed that nucleosomes can flip between the distal and proximal states, although a small proportion remained in either the distal (10%) or proximal (22%) states (Fig. 3b). To rule out the possibility that the observed dynamic FRET stems from movement of DNA, we relocated the donor to H2A (Extended Data Fig. 4e–g), and observed similar dynamic FRET transitions that showed the same slight preference for the dye-proximal dimer. Alternative explanations for flipping, such as DNA unwrapping or SWR1 diffusing along the DNA overhang, were ruled out because unwrapping[12] and diffusion[21] require ATP binding, whereas flipping is not dependent on ATP.

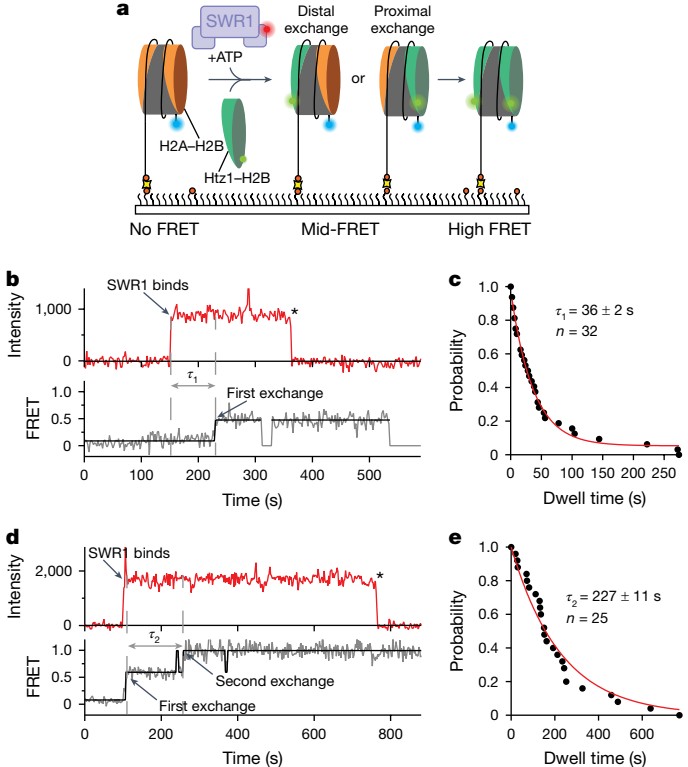

**Fig. 2 | SWR1 processively exchanges H2A–H2B for Htz1–H2B. a**, Three-colour smFRET assay with surface-immobilized AF488–nucleosome (blue), AF555–Htz1–H2B dimers (green) and SWR1(647N) (red). SWR1 binding is monitored by red fluorescence; histone exchange is detected as a FRET increase between AF488 and AF555. **b**, Example trace showing SWR1 binding (red; top) followed by a single-exchange event (bottom) after the exchange time ($\tau_1$). Asterisk indicates photobleaching or dissociation. **c**, Dwell time distribution between SWR1 binding and the first exchange (for both single and double exchanges) yields a $\tau_1 = 36 \pm 2$ s. **d**, Example trace showing SWR1 binding (red; top) followed by two processive exchange events (bottom) with a second-exchange time ($\tau_2$). Asterisk indicates photobleaching or dissociation. **e**, Dwell time distribution between the first and second exchanges yields a second-exchange time $\tau_2 = 227 \pm 11$ s. Reported errors are the error of the fit.

A time-binned FRET histogram of all the trajectories (Fig. 3c) showed that each state is sampled with approximately equal probability with a slight preference for the dye-proximal state. A dwell time analysis of the dynamic molecules showed that the nucleosome flips with approximately equal average time (3–4 s; Fig. 3d) from either the proximal or the distal face of the nucleosome. This observation is consistent with previous data from our group (and that in Fig. 1) that show an approximately equal propensity for exchange of each of the dimers in a yeast nucleosome in the first step[12]. The dwell time analysis further revealed biphasic exponential kinetics with a slow (4–5 s) and a fast (0.6–0.8 s) population, indicating the presence of flipping intermediates that cannot be distinguished by FRET alone (Extended Data Fig. 4b). A possible explanation for the biphasic kinetics is that SWR1 engages the nucleosome in either a more (slow) or less (fast) stable conformation. The corrected exponential amplitudes showed that the molecules spend approximately 80% of the time in the slow (more engaged) configuration.

## SWR1 senses heterotypic nucleosome dimer

Although each dimer in a canonical yeast nucleosome containing two H2A–H2B dimers has, in principle, an equal likelihood to be exchanged, the second-exchange reaction is exclusively of the remaining H2A-containing dimer[15]. Furthermore, a nucleosome in which both H2A–H2B dimers have been exchanged for Htz1–H2B cannot undergo SWR1-catalysed replacement by Htz1, and even futile cycling, in which one Htz1–H2B dimer is exchanged for another, does not seem to occur[15]. These observations indicate that the SWR1 complex has a mechanism to distinguish between Htz1 and H2A within a nucleosome.

Having determined that the bound nucleosome flips between conformations and that each presents a different nucleosome face, and hence dimer, to the SWR1 complex, we then sought to test whether this mechanism allowed SWR1 to probe the identity of the dimer with which it was presented. We prepared nucleosomes with a single copy each of H2A and Htz1 (Fig. 4a) and then repeated the experiments described above to monitor the flipping process. The data show that these 'heterotypic' nucleosomes bind to SWR1 in a similar manner to the canonical nucleosomes and are able to flip between both distal and proximal orientations (Fig. 4b and Extended Data Fig. 4h). However, in contrast to canonical nucleosomes, the static trajectories were almost exclusively in the proximal orientation that presents the H2A–H2B dimer to SWR1 (Fig. 4b). Furthermore, dwell time analysis for each orientation revealed clear kinetic differences between the two states (Fig. 4d and Extended Data Fig. 4i). The side containing the H2A–H2B dimer exhibits almost identical kinetics to the canonical nucleosome (complare with Fig. 3d). By contrast, the side containing the Htz1–H2B dimer exhibits only a single fast exponential decay (0.63 ± 0.02 s), comparable with the fast component of the canonical dimer face. This is also reflected in the time-binned FRET histogram (Fig. 4c), which shows a clear preference for the proximal (H2A–H2B) face.

To rule out the possibility that the observed kinetic differences stem from the asymmetric DNA overhangs, we prepared heterotypic nucleosomes with swapped DNA overhangs (Extended Data Fig. 4k–n). The data show that the observed biased flipping kinetics is maintained, confirming that the SWR1 selection against the Htz1–H2B side is based on histone content rather than on DNA overhang. These data show that SWR1 is able to distinguish between H2A and Htz1 within the context of the nucleosome, discriminating against Htz1 by rapidly flipping back to the canonical side. We thus propose that a form of kinetic proofreading[22,23] places the appropriate face of the nucleosome into the position proficient for dimer removal and exchange, thus contributing to the exquisite selectivity of the enzyme for replacing H2A with Htz1 and not the reverse. However, the ratio of the two dwell times (Fig. 4d) only gives a selectivity of sixfold, which is less than the apparent selectivity reported for SWR1 (refs. 15,16), so although this kinetic proofreading contributes significantly to specificity, additional steps (such as the multiple ATP hydrolysis events during dimer exchange or selective binding of Htz1–H2B versus H2A–H2B dimers for insertion) probably increase this selectivity further.

## The hexasome vacant site hinders flipping

A necessary intermediate in the histone dimer exchange reaction is a hexasome intermediate in which one H2A–H2B dimer has been removed but has not yet been replaced with an Htz1–H2B dimer. We next prepared hexasomes labelled in the same way as nucleosomes (Fig. 4e and Extended Data Fig. 5) to determine whether there was any effect on the kinetics and distribution of binding orientations. The single-molecule trajectories exhibit an almost complete loss of the proximal orientation with almost all hexasomes bound in the distal orientation, with the fraction of molecules flipping between the distal and the proximal sites decreasing to 14% (Fig. 4f). The distal orientation places the 'empty dimer' site against the SWR1 complex surface ready for insertion by an incoming Htz1–H2B dimer, consistent with structural data suggesting that this is the face of the nucleosome that undergoes histone exchange[12]. Although we did observe occasional flipping into the proximal conformation, this state is very short lived and reverts quickly to the distal configuration. A time-binned histogram

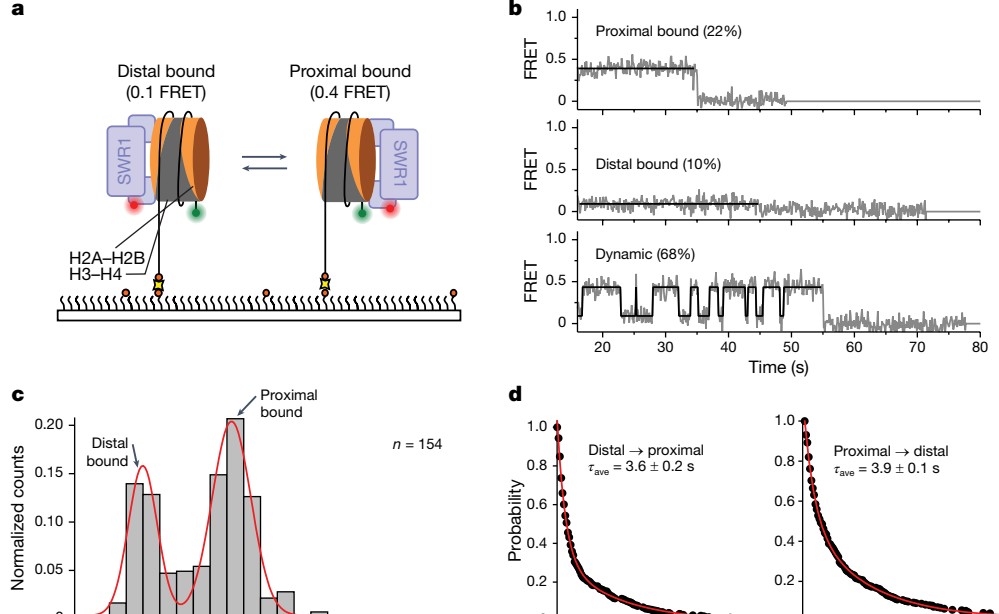

**Fig. 3 | SWR1 flips between each face of a nucleosome. a**, Schematic of the assay. Nucleosomes (113N2.Cy3) labelled with Cy3 on the short 2-bp overhang are surface immobilized on a PEGylated microscope slide. SWR1, labelled with Atto647N on the N terminus of the Arp6 subunit (SWR1(647N)) is flowed in and allowed to bind to the nucleosome. Interactions between the nucleosome and SWR1(647N) are monitored via smFRET between the donor (green circle) and acceptor (red circle). **b**, Examples of typical smFRET (grey) and idealized (black) traces. Some molecules display a static FRET of either 0.4 or 0.1, whereas other molecules dynamically flip between these two FRET states.

**c**, Idealized FRET histogram shows two major populations of SWR1(647N) bound to a nucleosome: a low-FRET (0.1) population corresponding to SWR1(647N) bound to the dye-distal side of the nucleosome, and a mid-FRET (approximately 0.4) population corresponding to SWR1(647N) bound to the dye-proximal side of the nucleosome. **d**, Dwell time plots for the distal-to-proximal (left) and proximal-to-distal (right) transition. The average dwell times ($\tau_{ave}$) for SWR1 bound in the distal and proximal orientations are approximately equal. Reported errors are the error of the fit.

of all trajectories confirms that the proximal orientation becomes almost undetectable (Fig. 4g). These results are consistent with SWR1 placing the nucleosome empty site in position ready to accept the incoming Htz1–H2B dimer. This change in the flipping dynamics suggests an active stabilization of the hexasome intermediate, produced on-enzyme, retaining the orientation that places the empty site in position to accept the incoming dimer. Indeed, our recent cryo-EM structure of the hexasome-bound SWR1 complex demonstrates that Swc5 has a role in complex stabilization[24].

## Structural basis for nucleosome flipping

To gain a better understanding of the flipping mechanism, we used cryo-EM analysis to examine different states of SWR1 complexed with nucleosomes. Our previous cryo-EM structures have shown one major state for the complex but also revealed several minor states, the function of which was not evident at that time[12]. However, in light of the new single-molecule data above, we re-examined these less-populated structural states in an expanded dataset to see whether these provided information about how nucleosomes might flip between different conformations. Further analysis and processing focused on these minor classes, revealing additional details (Extended Data Fig. 6). Two classes were of particular interest and resulted in structures at 3.8 Å and 4.7 Å, respectively (Fig. 5, Extended Data Table 1, Extended Data Figs. 6 and 7 and Supplementary Videos 1 and 2). One of these classes (described briefly in our previous work[12]) was very similar to the major structure, but a longer section of overhang DNA is evident that emanates from the lower gyre of the nucleosome and binds across the surface of SWR1 (configuration I; Fig. 5a,b and Supplementary Video 1). The DNA extending from the upper gyre is unwrapped from the nucleosome and binds to Arp6–Swc6 in the same manner as that

described for the main structure[12]. The second structure (configuration II; Fig. 5c,d and Supplementary Video 2) also showed a longer section of DNA overhang bound to SWR1, but this time, it was the DNA that extended from the upper gyre, which is released from Arp6–Swc6, that now binds across the same surface of SWR1 as the DNA from the lower gyre in configuration I. The DNA overhang from the lower gyre is released in this structure. Thus, the same DNA-binding surface on SWR1 binds different overhangs in each structure (Extended Data Figs. 7 and 8).

The improved resolution, combined with the availability of an AlphaFold model for Swc2 (ref. 25) allows us to assign, locate and build regions of the Swc2 subunit that we were previously unable to assign confidently. Swc2 has an essential role in SWR1-mediated histone exchange[12,26]. The N-terminal region of Swc2 binds to Htz1–H2B dimers[26,27]; however, we are still unable to assign that part of Swc2 in our structure. The central portion of yeast Swc2 (residues 136–345) is a DNA-binding module that probably localizes the SWR1 complex towards the nucleosome-depleted region[21,28] (Extended Data Fig. 8a). We can confidently build a portion of this DNA-binding region (residues 195–329) into the density (Extended Data Fig. 8b). Our structure indicates that there are three contacts between this region of Swc2 and the DNA (Extended Data Fig. 8b). Several positively charged residues in these regions are conserved across Swc2 subunits from different species (Extended Data Fig. 9a), consistent with a role in interacting with DNA. Two of these contact regions have been previously observed[12], although specific residue contacts could not be unambiguously determined. The AlphaFold model of Swc2 now allows us to better define these regions, which both contact the DNA wrap of the nucleosome (Extended Data Fig. 8b). A third contact region, involving eight conserved basic residues (K319–K322, R325, K326, K328 and K329), are in a loop that sits across the surface of the SWR1 complex (Extended Data

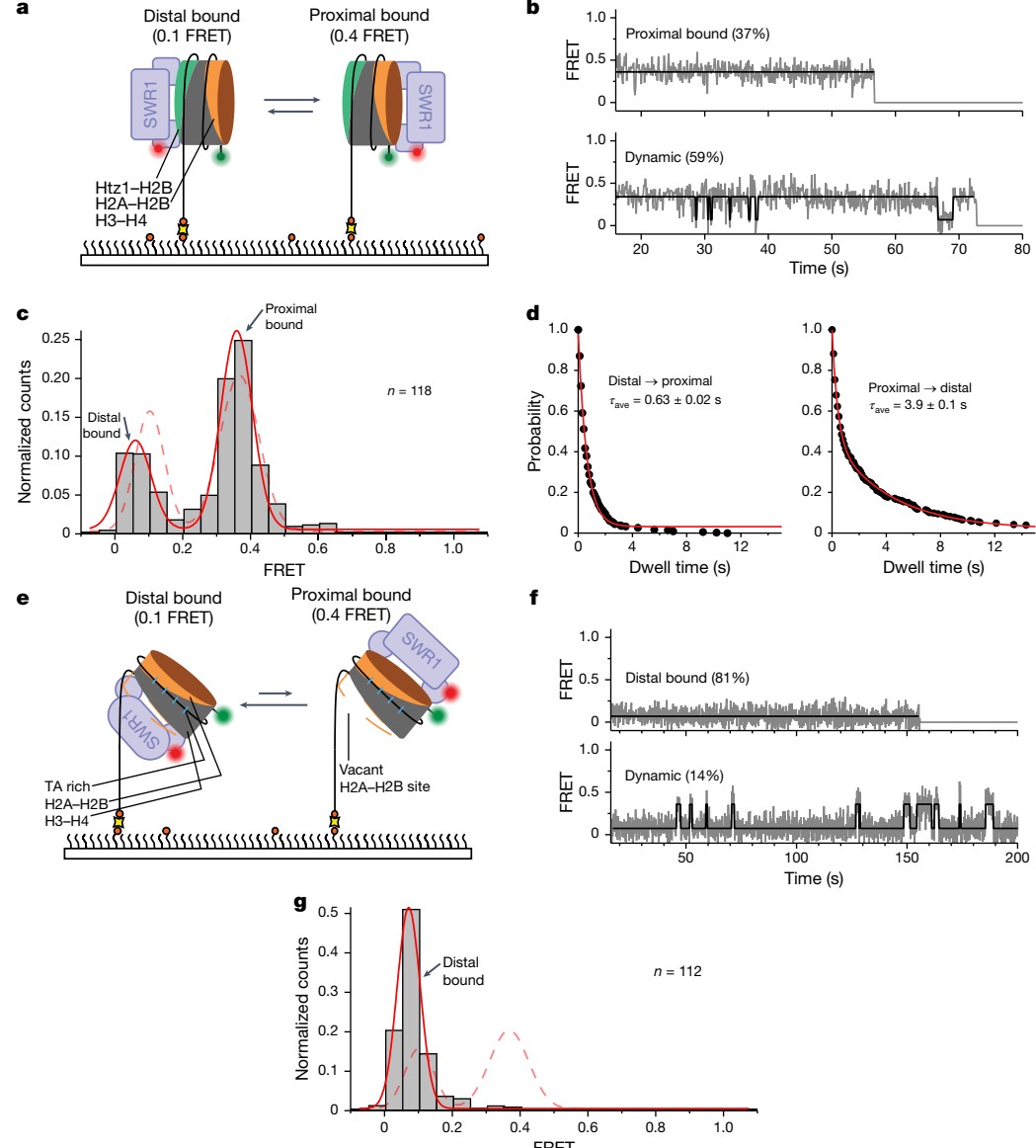

**Fig. 4 | Histone composition regulates SWR1 flipping kinetics. a**, Cy3-labelled surface-immobilized heterotypic nucleosomes (113N2.Cy3) containing Htz1–H2B (green) and canonical H2A–H2B dimers (orange). SWR1(647N) is flowed in and nucleosome binding is monitored via smFRET between the donor (green circle) and acceptor (red circle). **b**, Characteristic smFRET (grey) and idealized (black) trajectories. Some molecules (37%; $n_{total}$ = 118) display static 0.4 FRET, whereas others (59%) flip dynamically between 0.4 and 0.1 FRET. **c**, Idealized FRET histogram showing two populations corresponding to SWR1(647N) bound to the dye-distal Htz1–H2B (0.1 FRET) or to the dye-proximal H2A–H2B (approximately 0.4 FRET). The dashed line indicates the canonical nucleosome distribution from Fig. 3c. A small (0.04) shift of the low-FRET population may indicate altered binding to the Htz1 side. **d**, Dwell time plots for the distal-to-proximal (left) and proximal-to-distal (right) transition for a heterotypic nucleosome. Average dwell time ($\tau_{ave}$) on the Htz1–H2B side (distal) is shorter than the H2A–H2B side (proximal). **e**, Surface-immobilized hexasomes (113H2. Cy3) lacking the dye-distal H2A–H2B dimer (dashed orange line). SWR1(647N) is flowed in and hexasome binding is monitored via smFRET. **f**, Characteristic smFRET (grey) and idealized (black) trajectories. Molecules display a low (0.1) FRET. A small number of molecules show infrequent transitions from 0.1 to 0.4 FRET. **g**, Idealized FRET histogram showing one major population of SWR1(647N) bound to a hexasome. Only the low-FRET (0.1) population corresponding to SWR1(647N) bound to the vacant (dye-distal) side of the hexasome is present. The dashed line indicates the canonical nucleosome distribution from Fig. 3c. A small (0.03) shift of the low-FRET population may indicate altered binding when SWR1 faces the empty side. Reported errors are the error of the fit.

Figs. 8 and 9). This surface contacts the DNA overhang adjacent to the nucleosome wrap (Extended Data Fig. 8), but a different overhang in structures emanating from either the lower (configuration I) or the upper (configuration II) DNA gyre.

These two states suggest a simple mechanism to allow flipping of nucleosome orientations between the proximal and distal states (Fig. 5g). By swapping which DNA overhang is bound to SWR1, and then releasing the nucleosome but without releasing the DNA overhang, the nucleosome can flip and rebind in the opposite orientation but remain tethered to SWR1 by the DNA contact with the Swc2 subunit (Supplementary Video 3). Owing to the symmetry of the histone octamer, we cannot distinguish whether the nucleosome orientation relative to SWR1 switches between configurations I and II. An alternative explanation could be that SWR1 remains bound to the same face of the nucleosome in both configurations, and only the DNA overhang interacting with Swc2 is swapped. However, the smFRET experiments in which the donor is located on H2A still exhibit the nucleosome flipping dynamics (Extended Data Fig. 4e–g), thereby ruling out this possibility. Consequently, we

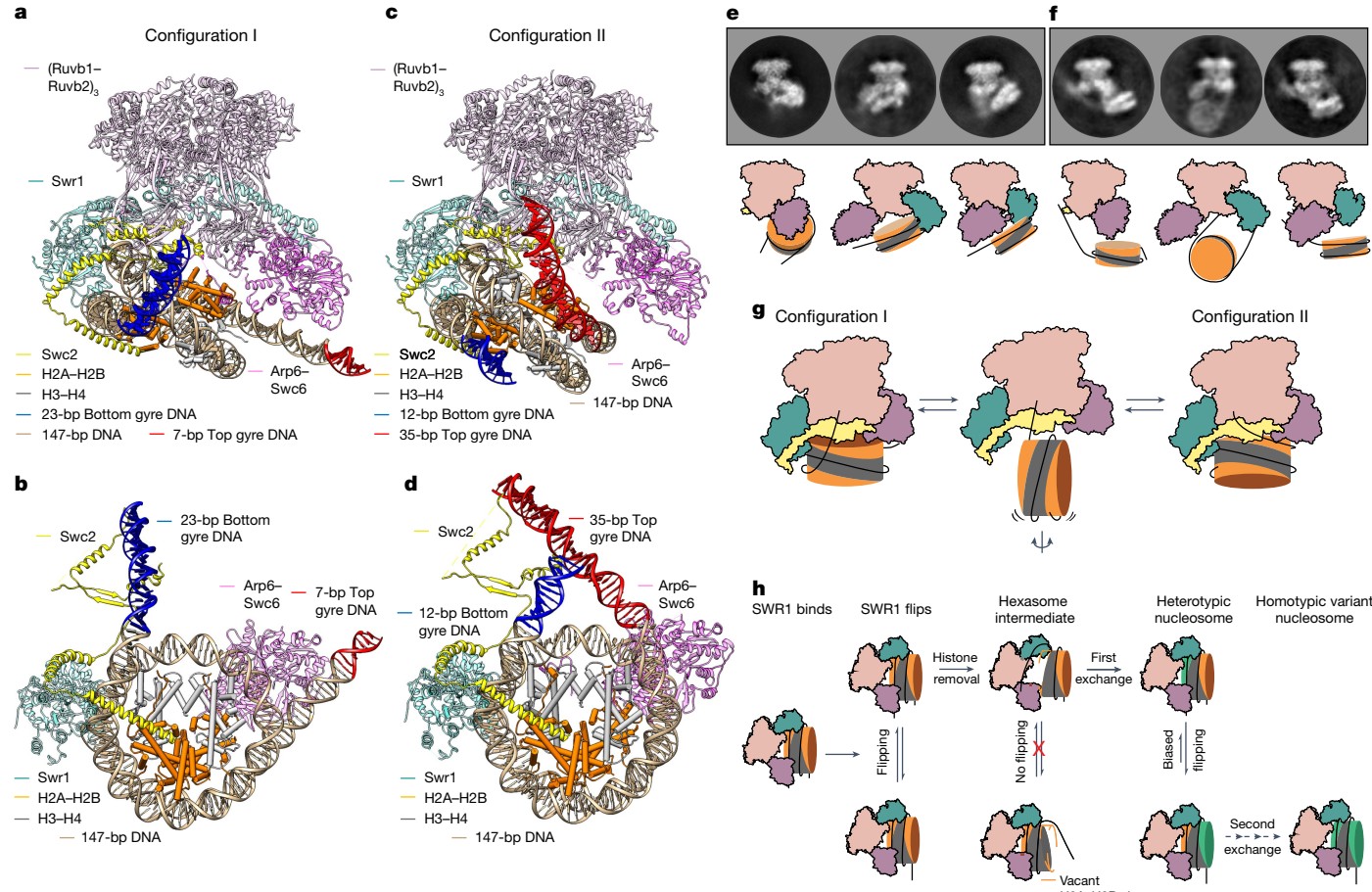

**Fig. 5 | Structural basis of SWR1-mediated nucleosome flipping. a**, The built-in coordinates of SWR1 in complex with a canonical nucleosome at 3.8 Å resolution in configuration I. Note that the DNA emanating from the lower gyre of the nucleosome (highlighted in blue) is bent up and binding across the surface of SWR1. **b**, A bottom view of the SWR1–nucleosome structure in configuration I. For clarity, only Swr1 (HD1 and HD2), Swc2 and the Arp6–Swc6 complex of SWR1 are shown. **c**, The built-in coordinates of SWR1 in complex with a canonical nucleosome at 4.7 Å resolution in configuration II. Note that the DNA emanating from the upper gyre of the nucleosome (highlighted in red) is binding across the surface of SWR1. **d**, A bottom view of the SWR1–nucleosome structure in configuration II. For clarity, only Swr1 (HD1 and HD2), Swc2 and the Arp6–Swc6 complex of SWR1 are shown. **e**, Three representative 2D class averages of SWR1–nucleosome in the canonical conformation. A cartoon representation of each 2D class is shown beneath. **f**, Three representative 2D class averages of SWR1-mediated nucleosome flipping with SWR1 orientated as in panel **e**. A cartoon representation of each 2D class is shown beneath. **g**, Cartoon summary of SWR1-mediated nucleosome flipping. **h**, Cartoon summary of kinetic proofreading and processivity of histone exchange by the SWR1 remodeller.

interpret that the two major states that we observed by cryo-EM represent intermediates on the flipping pathway. In principle, SWR1 could also use a single approximately 35-bp or longer DNA overhang while flipping the nucleosome (Fig. 5c), as shown in our single-molecule experiments (Fig. 3a). In the cell, however, the chromatin context (that is, the presence or absence of a neighbouring nucleosome) would dictate whether one or two overhangs can be used for flipping.

The single-molecule analysis presented above shows that although nucleosomes bound to SWR1 flip between configurations I and II, they only spend a very small amount of time flipping between these configurations. We would, therefore, expect to see very few complexes caught in this process in our cryo-EM dataset and these would probably be in various conformations with the nucleosome only interacting via the DNA overhangs that are so hard to define and average. Nonetheless, by careful classification of the particle dataset where bound nucleosome could not be visualized (Extended Data Fig. 6), we were able to identify several 2D classes in which the nucleosome could be observed in a flipped state where the nucleosome was disengaged from SWR1 but the flanking DNA remained bound. Three particularly well-defined examples are shown in Fig. 5 (compare Fig. 5e and 5f), but others were also

visible (Extended Data Fig. 9b). We interpret these as direct visualization of nucleosomes frozen in the act of flipping between configurations I and II. The dynamic nature of the flipping, however, precluded a 3D analysis of intermediate flipped states.

## Discussion

The ATP-dependent exchange of H2A–H2B dimers for those containing Htz1–H2B is a two-step process that replaces each dimer in turn[15]. Structural data[12] have strongly suggested that the dimer to be exchanged makes close contacts with SWR1, and unwrapping of the DNA from around this dimer begins upon binding and then progresses in an ATP-dependent process[18–20]. However, both dimers in a nucleosome can be exchanged, implying dissociation of the nucleosome to allow it to rebind with the appropriate dimer exposed for exchange in a distributive mechanism. On the basis of ensemble-averaged experiments, we have previously suggested that histone exchange proceeds via a distributive mechanism[12]. However, this conclusion was based on the assumptions that both exchanges proceeded with comparable rates, and that SWR1 would be completely processive or

distributive. The three-colour single-molecule exchange experiments (Fig. 2) showed that, under these conditions, the second exchange is much slower and that not all exchanges are processive, illustrating the need for such single-molecule experiments to unambiguously show processivity of the SWR1 complex. Why the second exchange is slower than the first one remains unclear. One possibility is that heterotypic nucleosomes have slower exchange rates. To test this possibility, we performed single-molecule exchange assays using a heterotypic nucleosome substrate (Extended Data Fig. 3). The resulting exchange time is approximately 100 s, still threefold slower than the first exchange (approximately 36 s), confirming that exchange is slower on heterotypic nucleosomes and in agreement with previous results[20,29].

However, demonstration that dimer exchange is processive raises a conundrum. We questioned how SWR1 accesses both faces of a single nucleosome without dissociation or by utilizing different mechanisms for each exchange given the asymmetric manner of association with the SWR1 complex. The answer is by a partial release of the nucleosome, while retaining a hold on the flanking DNA, to allow it to flip 180° and then rebind with the opposite face towards the enzyme to allow the second-exchange event (Fig. 5g,h and Supplementary Video 3). This simple, yet elegant, mechanism also explains the exquisite specificity of dimer exchange by SWR1 through dynamic, kinetic proofreading that favours placing a nucleosome face containing H2A rather than Htz1 in the position for exchange without releasing the substrate.

In cells, the Htz1–H2B variant is enriched at the +1 nucleosome at transcription start sites, which is typically flanked by a long (approximately 140 bp) nucleosome-free region (NFR) on one side[30]. SWR1 has been found, by crosslinking, to reside on the NFR-proximal side of the +1 nucleosome, which led the authors to question how SWR1 might exchange NFR-distal dimers[31]. At certain genes, chromatin immunoprecipitation-exo data from yeast cells have shown a preference for Htz1 insertion into the NFR-distal side of the nucleosome[32]; however, at the genome-wide level, the NFR-distal preference was closer to approximately 60:40, suggesting the opposite to be the case at other transcription start sites, despite the location of the SWR1 complex on the opposite face of the nucleosome[31]. In vitro, linker-distal or linker-proximal dimer preference for the first-exchange reaction has been somewhat controversial, as discussed[18]. Our initial studies[12] have shown a weak exchange preference (between 50:50 and 60:40) for the linker-distal (dye-proximal) dimer. Our new single-molecule exchange data (Fig. 1c) are consistent with those results (approximately 55:45 linker-distal). Data from other laboratories are also consistent with a weak linker-distal preference, particularly at physiological temperatures[29]. Conversely, others have reported a stronger preference for the linker-distal dimer based on more frequent and faster kinetics for linker-distal exchange[18,19]. The origin of these differences is not clear. The nature of the enzyme or histone source (for example, recombinant versus native, and yeast versus frog/*Drosophila*) or variations of the nucleosome positioning sequence could, in principle, explain these differences, but the data presented here do not resolve this issue.

Finally, we speculate that nucleosome flipping might have a role in other nucleosome remodelling activities. Flipping could be used to recognize and modify different histone components on both faces of nucleosomes or to monitor the histone composition of different faces of nucleosomes as part of regulation processes. Furthermore, the sliding directionality of nucleosomes on DNA could be swapped by such a flipping mechanism, as proposed for Chd1 (ref. 33), allowing processive sliding of nucleosomes to space them evenly along DNA or to position them in gene regulation and/or DNA repair.

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

## Methods

### Purification of wild-type SWR1

Recombinant SWR1 was produced as previously described[12,16] with minor modifications. Baculoviruses encoding SWR1 genes were initially amplified in Sf9 cells, before using the amplified baculoviruses to infect BTI-TN-5B1-4 (High Five) cells for expression, which were harvested after 72 h. Cells were lysed by sonication in 50 mM HEPES (pH 8.0), 0.5 M NaCl, 1 mM TCEP, 10% glycerol, 1 mM benzamidine-HCL supplemented with 1 protease inhibitor tablet and 10 µl of benzonase per litre of cell culture. Lysate was clarified by centrifugation at 30,000$g$ for 60 min at 4 °C. The supernatant was filtered before being injected onto a StrepTrap HP (Cytiva) column. The column was washed with buffer A (25 mM HEPES (pH 7.5), 0.3 M NaCl, 1 mM TCEP and 10% glycerol) before being eluted with buffer A supplemented with 5 mM desthiobiotin. The eluted protein was combined and diluted 1:1 with buffer B (25 mM HEPES (pH 7.5), 0.1 M NaCl, 1 mM TCEP and 10% glycerol) to dilute the salt before being loaded onto a HiTrap Q HP (Cytiva) column. The protein was eluted with a linear gradient from buffer B to buffer C (25 mM HEPES (pH 7.5), 2 M NaCl, 1 mM TCEP and 10% glycerol). The relevant fractions were pooled and diluted again 1:1 with buffer B to reduce the salt before being injected onto a Heparin HP (Cytiva) column. Protein was eluted with a linear gradient from buffer B to buffer C. Finally, the protein was concentrated, snap frozen in liquid nitrogen and stored at −80 °C.

### Purification of fluorescently labelled SWR1

To site specifically label the SWR1 complex, we made use of the ybbR-labelling approach[34,35]. The 11-amino acid ybbR tag was fused to the N terminus of the Arp6 subunit of SWR1. The ybbR–Arp6 mutant was used in place of the wild-type *Arp6* gene when assembling the SWR1 genes using the MultiBac system[16]. The SWR1(ybbR–Arp6) complex was expressed and purified in an analogous way to wild-type SWR1 with the ybbR-labelling reaction taking place after elution from the HiTrap Q HP column. The labelling reaction was carried out overnight at 4 °C. Typically, SWR1(ybbR–Arp6; approximately 1 µM) was labelled with CoA-Atto647N (approximately 10 µM) using recombinant Sfp transferase (approximately 0.2 µM) in buffer B supplemented with 10 mM MgCl$_2$. The labelled SWR1 complex was separated from free dye and Sfp transferase using a Heparin HP (Cytiva) column, eluting with a linear gradient from buffer B to buffer C. Finally, SWR1(Atto647N–Arp6) (referred to as SWR1(647N) in the text) was concentrated, snap frozen in liquid nitrogen and stored at −80 °C.

### Purification of *S. cerevisiae* histones

All nucleosomes or hexasomes used in this study were composed of *S. cerevisiae* histones assembled on DNA containing the 601 Widom sequence.

*S. cerevisiae* octamers with and without Alexa Fluor 555 on H2A K119C were prepared as previously described[16].

*S. cerevisiae* H2A–H2B, Htz1–H2B (with and without Alexa Fluor 555 on Htz1 K125C) or Htz1–H2B(3×Flag) histone dimers were expressed in *E. coli* and purified as soluble dimers. Cells were lysed by sonication in buffer D (20 mM Tris (pH 7.5), 0.5 M NaCl, 0.1 mM EDTA and 1 mM TCEP) plus protease inhibitor tablets (Roche; 2 tablets per 100 ml). Dimers were purified by loading the cleared lysate onto tandem HiTrap Q FF and HiTrap Heparin HP columns in buffer E (20 mM Tris (pH 7.5), 0.5 M NaCl, 1 mM EDTA and 1 mM TCEP). The HiTrap Q FF column was removed before elution from the HiTrap Heparin HP column via a gradient to buffer F (20 mM Tris (pH 7.5), 2 M NaCl, 1 mM EDTA and 1 mM TCEP), followed by gel filtration on a Superdex S200 in buffer F.

*S. cerevisiae* histone H3(Q120M, K121P and K125Q) and histone H4 were co-expressed in *E. coli* and purified as soluble tetramers. Cells were lysed by sonication in buffer D plus protease inhibitor tablets (Roche; 2 tablets per 100 ml). Tetramers were purified using a HiTrap Heparin HP column in buffer E and eluted via a gradient to buffer F, followed by gel filtration on a Superdex S200 in buffer F.

### Preparation of nucleosomes

Biotinylated DNA containing the Widom 601 sequence was generated as previously described[12]. Salt gradient dialysis of the *S. cerevisiae* octamers with DNA was carried out to form a 'core' nucleosome. A biotinylated DNA overhang was ligated to the core nucleosome as previously described[12]. This resulted in nucleosomes with one long overhang of 113 bp and a short overhang of 2 bp, which we refer to as 113N2 ('N' representing the Widom 601 nucleosome positioning sequence). The biotin was present on the long 113-bp linker. For nucleosomes where the DNA was labelled, the fluorophore was attached at the end of the 2-bp short overhang.

### Preparation of hexasomes

To facilitate the formation of yeast hexasomes, three amino acid substitutions were introduced into the *S. cerevisiae* H3 histone (Q120M, K121P and K125Q)[36]. These substitutions (MPQ) are the corresponding amino acids found in human and *Xenopus laevis* H3.

To form hexasomes, *S. cerevisiae* H2A–H2B dimers were mixed with *S. cerevisiae* H3(MPQ)–H4 tetramers. The amount of H2A–H2B dimers used was limited to 0.6× the amount of tetramers to ensure only partial H2A–H2B occupancy. Hexasomes were assembled onto the same DNA that was used for nucleosomes by salt gradient dialysis to generate 'core' hexasomes. Core hexasomes were separated from tetrasomes, nucleosomes and free DNA using a MonoQ column, loaded in buffer G (20 mM Tris (pH 7.5), 1 mM EDTA, 1 mM TCEP and 200 mM NaCl) eluting with a gradient into buffer H (as buffer G with 2 M NaCl). The fractions were immediately diluted into 4× volume of 20 mM Tris (pH 7.5) to reduce the salt concentration. A biotinylated DNA overhang was ligated to the core hexasome in the same way as was used for nucleosomes. This resulted in a hexasome with one long overhang of 113 bp and a short overhang of 2 bp, which we refer to as 113H2 ('H' representing a hexasome assembled on the Widom 601 sequence). The biotin was present on the long 113-bp linker. For hexasomes where the DNA was labelled, the fluorophore was attached at the end of the 2-bp short overhang.

As is the case for hexasomes prepared with *X. laevis* histones[37], yeast hexasomes prepared in this way exploit the inherent asymmetry of the Widom 601 sequence. Because of this asymmetry, the H2A–H2B dimer present in a hexasome is preferentially located on the 'TA-rich' side of the Widom 601 sequence, leaving the vacant site on the 'TA-poor' side. We orientated our Widom 601 sequence with the TA-rich side closest to the 2-bp short overhang. This resulted in the vacant H2A–H2B site being located next to the 113-bp linker.

### Preparation of heterotypic nucleosomes

Core hexasomes, prepared as described above, were mixed with *S. cerevisiae* Htz1–H2B dimers to form heterotypic nucleosomes. Htz1–H2B dimers were added at an amount equal to 0.3× the amount of hexasome present. Core heterotypic nucleosomes were then purified in the same way as canonical nucleosomes. A biotinylated DNA overhang was ligated to the core heterotypic nucleosomes as described above. Resulting heterotypic nucleosomes contain the Htz1–H2B dimer next to the long 113-bp overhang and the conical H2A–H2B dimer next to the short 2-bp overhang.

### Bulk histone exchange assay

SWR1 (100 nM; wild type or SWR1(647N)), 200 nM nucleosomes and 400 nM Htz1–H2B(3×Flag) were mixed in exchange buffer (25 mM Tris-HCl (pH 7.8), 100 mM KCl, 0.2 mM EDTA and 2 mM MgCl$_2$), with or without 1 mM ATP. The exchange reaction was carried out at 30 °C. At the indicated time points, 8 µl of the reaction was removed and quenched by the addition of 4 µl of a stopping solution (0.5 mg ml$^{-1}$ salmon sperm DNA, 30 mM EDTA and 3× ficoll loading buffer) and

placed on ice. The 'no ATP' control was taken at the longest indicated time point. After all time points had been taken, the reaction products were separated by 6% native PAGE, run at 110 V in 0.5× TBE at 4 °C and visualized using fluorescence of the nucleosome.

## Two-colour smFRET microscope

smFRET measurements looking at the flipping of nucleosomes by SWR1 were performed on an Olympus IX-71 microscope equipped with a homebuilt prism-TIRF module. Excitation was provided by a 532-nm laser (Stradus, Vortran) or a 637-nm laser (Stradus, Vortran). Fluorescence was collected through a 1.2 NA, 60× water objective (Olympus) and filtered through a dual bandpass filter (FF01-577/690-25, Semrock). The fluorescence was spectrally separated using a Opto-Split II (Cairn Research) to separate donor and acceptor emission. The donor and acceptor emissions were further filtered through ET585/65M and ET700/75M (Chroma) bandpass filters, respectively. The donor and acceptor images were then projected side-by-side onto an electron-multiplying charge-coupled device (EMCCD) (Andor iXon Ultra 897). Data were collected as raw movies using a custom LabVIEW script.

Single-molecule fluorescence spots from the raw movies were localized using custom IDL scripts and converted into raw fluorescence trajectories. Raw fluorescence trajectories were corrected for bleed through of the donor fluorescence into the acceptor channel. Apparent FRET efficiencies were calculated as the ratio of acceptor intensity divided by the sum of the donor and acceptor intensities.

Two mechanical shutters (LS-3, Uniblitz, Vincent Associates) were placed in the excitation path for alternating laser excitation (Extended Data Fig. 4e–g). Frame acquisition and shutter synchronization were obtained using a homebuilt negative-edge-triggered JK flip–flop circuit (SN74LS112AN, Texas Instruments) using the 'Fire' output of the EMCCD as the input clock. IDL scripts were modified accordingly to locate single molecules and extract fluorescence trajectories.

## Three-colour smFRET microscope

smFRET measurements looking at histone exchange coupled with SWR1 binding were performed on an Olympus IX-71 microscope equipped with a homebuilt prism-TIRF module. Alternating laser excitation was provided by a 488-nm laser (OBIS, Coherent) or a 637-nm laser (OBIS, Coherent). Alternation of the lasers and synchronization of the lasers with the camera were controlled by a custom LabVIEW script and a DAQ (USB-6341, National Instruments). Fluorescence was collected through a 1.2 NA, 60× water objective (Olympus) and filtered through ET500lp (Chroma) and NF03-642E-25 (Semrock) filters. The fluorescence was spectrally separated using a MultiSplit (Cairn Research) housing the following dichroic filters: T500lpxr UF2, T635lpxr UF2 and T725lpxr UF2 (Chroma). The separated fluorescent emission was projected onto quadrants of a sCMOS (ORCA Fusion, Hammamatsu) camera. Data were collected as raw movies using HCImage Live (Hammamatsu).

Single-molecule fluorescence spots from the raw movies were localized using custom IDL scripts and converted into raw fluorescence trajectories. Raw fluorescence trajectories were corrected for bleed through of the donor fluorescence into the acceptor channel. Apparent FRET efficiencies were calculated as the ratio of acceptor intensity divided by the sum of the donor and acceptor intensities.

## Microscope slide passivation and flow chamber assembly

Quartz slides (UQC optics) and glass coverslips were aminosilinized with N-(2-aminoethyl)-3-aminopropyltrimethoxysilane, then passivated using methoxy-PEG-SVA (relative molecular mass = 5,000; Laysan Bio, Inc.) containing 5% biotin-PEG-SVA (relative molecular mass = 5,000, Laysan Bio, Inc.) in 100 mM sodium bicarbonate as previously described[38] with minor modifications. Following passivation, slides and coverslips were stored under nitrogen in the dark at −20 °C. Before use, slides and coverslips were warmed to room temperature

and assembled into flow chambers using 0.12-mm thick double-sided adhesive sheets (Grace Bio-Labs SecureSeal). Flow chambers were sealed with epoxy glue.

## Nucleosome or hexasome immobilization

Nucleosomes or hexasomes were surface immobilized as previously described[12]. In brief, neutravidin (0.1 mg ml$^{-1}$) in T50 buffer (50 mM Tris-HCl (pH 7.5) and 50 mM NaCl) was injected into the assembled flow chamber and incubated for 5 min to allow binding to the biotinylated PEG surface. Excess neutravidin was washed out with reaction buffer (25 mM Tris-HCl (pH 7.8), 100 mM KCl, 4% glycerol, 1 mM EDTA, 2 mM MgCl$_2$ and 0.2 mg ml$^{-1}$ BSA). Biotinylated nucleosomes or hexasomes were diluted to 10 pM in reaction buffer before injecting into the flow chamber and allowed to bind to the neutravidin for 5 min. Excess nucleosomes or hexasomes were flushed out using imaging buffer (reaction buffer with Trolox, 2.5 mM protocatechuic acid and 0.25 μM protocatechuate-3,4-dioxygenase) and imaged immediately.

## smFRET between nucleosome or hexasome and SWR1 data collection

Nucleosomes or hexasomes labelled with a Cy3 donor on the short end of the DNA overhang (113N2.Cy3 or 113H2.Cy3) were immobilized in a flow chamber and imaged. SWR1(647N), 10 nM in imaging buffer (25 mM Tris-HCl (pH 7.8), 100 mM KCl, 4% glycerol, 1 mM EDTA, 2 mM MgCl$_2$, 0.2 mg ml$^{-1}$ BSA, Trolox, 2.5 mM protocatechuic acid and 0.25 μM protocatechuate-3,4-dioxygenase) was injected. Imaging was performed by first directly exciting the acceptor with a 637-nm laser for approximately 15 s to localize SWR1(647N), before switching to 532-nm excitation to observe FRET between the nucleosome or hexasome and SWR1. All single-molecule measurements were carried out at room temperature, data were acquired with a 100-ms frame time.

## smFRET between nucleosome or hexasome and SWR1 data analysis

Manual inspection of the donor intensity, acceptor intensity and apparent FRET from each molecule was carried out using custom MATLAB scripts. For a molecule to be included in downstream analysis, it needed to have a constant signal from the acceptor under direct acceptor excitation to indicate that SWR1(647N) was bound and display a single step photobleaching event of either the donor or acceptor under donor excitation. All molecules that satisfied these criteria were truncated to just the FRETing region preceding the photobleaching event.

Truncated FRET traces were analysed with a hidden Markov model using vbFRET, using default parameters[39]. The idealized FRET from vbFRET was used to generate FRET histograms, plotted using Igor Pro 8 (Wavemetrics). Dwell times from the idealized FRET trajectories were extracted using custom MATLAB scripts. Only dwell times in which the idealized FRET transitioned between proximal and distal states (or the reverse) were included. Dwell time plots were generated in MATLAB and plotted in Igor Pro 8. The lifetime of the proximal-bound and distal-bound states was determined by fitting the dwell time plots to a double exponential function in Igor Pro 8. The slow and fast exponential phases probably correspond to a fully or partially engaged SWR1 complex, respectively. The average lifetimes ($\tau_{ave}$) for proximal-bound and distal-bound states were calculated using the pre-exponential factors ($A$) and lifetimes ($\tau$) determined from the double exponential fit as follows:

$$\tau_{ave} = (A_1\tau_1^2 + A_2\tau_2^2)/(A_1\tau_1 + A_2\tau_2)$$

In all cases, we observed both static and dynamic trajectories when probing the FRET between nucleosomes or hexasomes and SWR1. Only dynamic trajectories were used for determining the kinetics. For both the canonical and the heterotypic nucleosomes, static trajectories represent a minority of the observed molecules. Short

static traces may be due to dye photobleaching or SWR1 diffusion before a flipping event can take place. However, longer static traces are also observed. This heterogeneity is summarized in Extended Data Fig. 5. Long static trajectories suggest that a proportion of SWR1 molecules are stably engaged on one side of the nucleosome and not dynamically checking the histone identity of each nucleosome face. The nature of this stable SWR1 binding, compared with binding that allows nucleosome flipping, is unknown, as is the method by which SWR1 could transition from a static (stable binding) to a flipping (checking histone identity) state.

### smFRET real-time imaging of histone exchange and SWR1-binding data collection

A quartz flow cell was prepared as described above. Neutravidin (0.01 mg ml$^{-1}$) in T50 buffer (50 mM Tris-HCl (pH 7.5) and 50 mM NaCl) was injected into the flow chamber and incubated for 5 min to allow binding to the biotinylated PEG surface. Excess neutravidin was washed out and the flow cell further passivated by incubation with Pluronic F127 (0.5% w/v) in T50 buffer. Excess Pluronic F127 was washed out with reaction buffer (25 mM Tris-HCl (pH 7.8), 100 mM KCl, 4% glycerol, 0.2 mM EDTA, 2 mM MgCl$_2$ and 0.2 mg ml$^{-1}$ BSA).

To follow the insertion of variant histones in real time at the single-molecule level, a 'gain of FRET' assay was used. Nucleosomes labelled with Alexa Fluor 488 (FRET donor) on the short 2-bp overhang (113N2.AF488) were immobilized in a flow chamber and imaged. To start the reaction, 1 nM SWR1, 4 nM Chz1–Htz1(AF555)–H2B and 1 mM ATP in imaging buffer (25 mM Tris-HCl (pH 7.8), 100 mM KCl, 4% glycerol, 0.2 mM EDTA, 2 mM MgCl$_2$, 0.2 mg ml$^{-1}$ BSA, Trolox, 2.5 mM protocatechuic acid and 0.25 µM protocatechuate-3,4-dioxygenase) was injected into the chamber using a syringe pump. Exchange can be monitored by stepwise FRET increases as the AF555-labelled (FRET acceptor) Htz1–H2B dimer is exchanged into the immobilized AF488-labelled nucleosome. To reduce nonspecific binding of the Htz1(AF555)–H2B dimer, the dimer was first complexed with its natural chaperone, Chz1 (ref. 40).

For experiments that simultaneously followed exchange and SWR1 binding, the experiment was conducted as described but with SWR1(647N) using the three-colour smFRET microscope described above. The two excitation lasers (488 nm and 637 nm) were alternated at a frequency of 1 Hz. All experiments were carried out at room temperature (22 °C).

### smFRET real-time imaging of histone exchange and SWR1-binding data analysis

Visualization of single-molecule trajectories was carried out using custom MATLAB scripts. For each single molecule, the intensity of the donor (Alexa Fluor 488), acceptor (Alexa Fluor 555) and corresponding FRET, along with the colocalized SWR1-binding intensity (Atto647N) were inspected. Nucleosomes that underwent exchange were identified by stepwise increases in the FRET trajectory. SWR1 binding was identified as an increase in the Atto647N intensity. Nucleosomes where the signal for SWR1 binding overlapped with at least one exchange event were included for further analysis. Dwell times were collected by manual inspection of the trajectories. Data were obtained by measuring several regions of interest from at least three independent slides. Dwell time plots were generated in MATLAB and plotted and fit in Igor Pro 8.

### Single-molecule measurements of SWR1 nucleosome lifetime

Nucleosomes labelled with Alexa Fluor 488 on the short 2-bp overhang (113N2.AF488) were immobilized in a flow chamber as described above. Of SWR1(647N), 5 nM in imaging buffer (25 mM Tris-HCl (pH 7.8), 100 mM KCl, 4% glycerol, 1 mM EDTA, 2 mM MgCl$_2$, 0.2 mg ml$^{-1}$ BSA, Trolox, 2.5 mM protocatechuic acid and 0.25 µM protocatechuate-3,4-dioxygenase with 1 mM ATP) was injected. The three-colour smFRET microscope described above was used. The two excitation lasers (488 nm and 637 nm) were alternated at a

frequency of 1 Hz. Experiments were carried out at room temperature (22 °C). Trajectories in which SWR1(647N) colocalized with a nucleosome were selected and further processed using tMAVEN[41] to determine the time for SWR1 to bind and the time SWR1 remained bound to a nucleosome.

### Preparation of the SWR1–nucleosome complex for cryo-EM

Recombinant SWR1 was produced in BTI-TN-5B1-4 (High Five) insect cells, and the SWR1–nucleosome complex was assembled as previously described[12]. SWR1–nucleosome grids were prepared as previously described, except instead of glow discharge, the grids were cleaned by washing with water and ethyl acetate. Cryo-EM data acquisition, image acquisition and structure reconstruction were conducted using a similar procedure as previously described[12]. Data processing and refinement statistics for the two cryo-EM structures are summarized in Extended Data Table 1.

### Cryo-EM data collection

A total of 35,076 micrographs were collected using a Titan KRIOS microscope operated at 300 kV. Images were collected on a Falcon IV direct electron detector with a pixel size of 1.1 Å px$^{-1}$. Images were collected with a defocus range of −0.7 to −1.9 µm, with 1.0 s exposure time and a total dose of 40 e$^-$ Å$^{-2}$ fractionated over 39 frames.

### Cryo-EM data processing

Movie frames were aligned using MotionCor2 (ref. 42), as previously described[12]. Contrast transfer function parameters were determined using Gctf[43] as previously described[12]. Particle picking was performed in cryoSPARC[44], as previously described[12]. Global-resolution and local-resolution estimates were calculated based on the gold-standard Fourier shell correlation (FSC = 0.143) criterion.

The cryo-EM processing workflow for the 3.8 Å SWR1–nucleosome map in configuration I is summarized in Extended Data Fig. 6. First, in the recently collected SWR1–nucleosome dataset, 2D classification in cryoSPARC for 2D classes containing density for SWR1 or the nucleosome resulted in a working particle pool of 1,918,312 particles[44]. These were subdivided into three classes via heterogeneous refinement in cryoSPARC, resulting in class 1 (SWR1–nucleosome complex (15%)), class 2 (SWR1-apo (55%)) and class 3 (nucleosome only (30%)). The subset of 268,805 particles in class 1 (SWR1–nucleosome) was then further classified into five classes via heterogeneous refinement in cryoSPARC, resulting in class 1.1 (SWR1–nucleosome in configuration I (68%)), class 1.2 (SWR1–nucleosome configuration II (17%)), class 1.3 (poorly aligned class (9%)), class 1.4 (poorly aligned class (2%)) and class 1.5 (poorly aligned class (4%)). The particles in class 1.1 were then imported and subjected to 3D refinement in RELION before one round of 3D classification without alignment ($T$ = 30), with a soft mask overlapping the Swc2–bottom gyre DNA interface[45]. This generated two classes: class 1.1.1 (no density for bottom gyre DNA (63%)) and class 1.1.2 (clear density for bottom gyre DNA (37%)). Particles in class 1.1.2 were further selected for 3D refinement in RELION.

Next, in the previously collected dataset, 2D classification in cryoSPARC for 2D classes containing density for SWR1 or the nucleosome resulted in a working particle pool of 296,061 particles. These were subdivided into three classes via heterogeneous refinement in cryoSPARC, resulting in a class 1.1 (SWR1–nucleosome complex (33%)), class 1.2 (SWR1-apo (39%)) and class 1.3 (nucleosome only (28%))[44]. The subset of 96,648 SWR1–nucleosome particles were then further classified into five classes via heterogeneous refinement in cryoSPARC, resulting in class 1.1 (SWR1–nucleosome in configuration I (68%)), class 1.2 (SWR1–nucleosome configuration II (23%)), class 1.3 (poorly aligned class (5%)), class 1.4 (poorly aligned class (2%)) and class 1.5 (poorly aligned class (2%)). Particles in class 1.1 were imported and refined in RELION before one round of 3D classification without alignment ($T$ = 30), with a soft mask overlapping the Swc2–bottom gyre DNA interface. This generated

two classes: class 1.1.1 (no density for bottom gyre DNA (16%)) and class 1.1.2 (clear density for bottom density (84%)). Particles in class 1.1.2 were further selected for 3D refinement in RELION[45]. Particles from classes 1.1.2 in the recently collected dataset and 1.1.2 in the previously collected dataset were then merged to generate a working pool of 123,591 particles. The resulting particles were then subjected to 3D refinement and contrast transfer function refinement in RELION with a mask corresponding to the SWR1 subcomplex of Swr1, Arp6, Swc6, Swc2, RuvBL1 and RuvBL2, and the nucleosome to generate the final 3.8 Å SWR1–nucleosome map in configuration I[45].

The cryo-EM processing workflow for the 4.7 Å SWR1–nucleosome map in configuration II is summarized in Extended Data Fig. 6. First, in the recently collected SWR1–nucleosome dataset, particles in class 1.2 were selected, generating a working pool of 35,102 particles. The subset of particles was further classified into two classes in RELION using 3D classification with alignment ($T = 6$) in the absence of a mask[45]. This generated class 1.2.1 (SWR1–nucleosome with poor density for the upper gyre DNA (39%)) and class 1.2.2 (SWR1–nucleosome with clearer density of upper gyre DNA (61%)). The particles in class 1.2.2 were selected, generating a working pool of 20,990 particles for 3D refinement in RELION.

Next, in the previously collected SWR1–nucleosome dataset, particles in class 1.2 were selected, generating a working pool of 21,054 particles. The subset of particles was further classified in two classes in RELION using 3D classification with alignment ($T = 6$) in the absence of a mask[45]. This generated class 1.2.1 (SWR1–nucleosome with poor density for the upper gyre DNA (40%)) and class 1.2.2 (SWR1–nucleosome with clearer density of upper gyre DNA (60%)). The particles in class 1.2.2 were selected, generating a working pool of 12,605 particles for 3D refinement in RELION[45]. Particles from classes 1.2.2 in the recently collected dataset and 1.2.2 in the previously collected dataset were then merged to generate a working pool of 33,595 particles. The resulting particles were then subjected to 3D refinement and contrast transfer function refinement in RELION with a mask corresponding to the SWR1 subcomplex of Swr1, Arp6, Swc6, Swc2, RuvBL1 and RuvBL2, and the nucleosome to generate the final 4.7 Å SWR1–nucleosome map in configuration II.

### Model building

For the Swc2 subunit, an initial template was generated using AlphaFold[25]. Different regions corresponding to secondary structures of the template were manually truncated and docked separately into the recently generated 3.8 Å SWR1–nucleosome map in configuration I in Chimera[12,46], before being further built in Coot[47]. The final coordinates were subjected to real-space refinement in Phenix[48].

For the 3.8 Å SWR1–nucleosome configuration I map, first the SWR1–nucleosome complex from the previously solved 3.6 Å SWR1–nucleosome structure (Protein Data Bank (PDB) ID 6GEJ) was docked into the density using Chimera[12,46]. The coordinates for the DNA were then omitted. Next, the SWR1–nucleosome complex from the previously solved 4.5 Å SWR1–nucleosome structure (PDB ID 6GEN) was superimposed onto the docked structure using RuvBL1 and RuvBL2 as a reference. Coordinates for the superimposed structure were then omitted, with exception to the coordinates for the DNA, which was kept and docked into the 3.8 Å SWR1–nucleosome configuration I map in Chimera, before merging the two PDB models: SWR1–nucleosome DNA omitted and DNA only together. The coordinates corresponding to the previously built Swc2 subunit were then omitted, and the coordinates for the newly built Swc2 model were docked into the map. Additional DNA overhang was then built manually in Coot[12,46,47]. The final coordinates were then subjected to real-space refinement in Phenix[48].

For the 4.7 Å SWR1–nucleosome configuration II map, SWR1 from the previously solved 3.6 Å SWR1–nucleosome structure (PDB ID 6GEJ) was docked into the density using Chimera[46]. The coordinates corresponding to Swc2 were omitted, and the recently built Swc2 was docked together into the density using Chimera and further built in Coot[46,47]. The additional DNA overhang was then built manually in Coot. The final coordinates were then subjected to real-space refinement in Phenix[48].

### 2D classification of SWR1-mediated nucleosome flipping

First, in the recently collected SWR1–nucleosome dataset, particles in class 2 (SWR1-apo (55%)) were selected, generating a working pool of 594,100 particles. The subset of particles was then further classified into four classes via heterogeneous refinement in cryoSPARC, resulting in class 2.1 (RuvBL1–RuvBL2 only (21%)), class 2.2 (a poorly aligned class (20%)), class 2.3 (SWR1-apo with additional density underneath SWR1 (38%)) and class 2.4 (a poorly aligned class (21%)). Particles in class 2.3 were then selected for 2D classification in RELION[45].

Next, in the previously collected SWR1–nucleosome dataset, particles in class 2 (SWR1-apo (39%)) were selected, generating a working pool of 115,463 particles. The subset of particles was then further classified into four classes via heterogeneous refinement in cryoSPARC, resulting in class 2.1 (RuvBL1–RuvBL2 only (25%)), class 2.2 (a poorly aligned class (20%)), class 2.3 (SWR1-apo with additional density underneath SWR1 (30%)) and class 2.4 (a poorly aligned class (25%)). Particles in class 2.3 were then selected for 2D classification in RELION. The particles in class 2.3 in the recently collected SWR1–nucleosome dataset and the particles in class 2.3 in the previously collected SWR1–nucleosome dataset were then merged and subjected to multiple rounds of 2D classification in RELION to obtain 2D classes of SWR1-mediated nucleosome flipping.

### Statistics and reproducibility

For data relating to Fig. 1, the total number of traces used in each dataset is indicated in each panel and was derived from three independent experiments. For data relating to Fig. 2, the total number of traces used for each dataset is indicated in each panel and was derived from four independent experiments. For data relating to Figs. 3 and 4, two independent experiments were performed, one of which is shown. The total number of traces used for each dataset is indicated in each panel. All gels were independently and successfully repeated twice.

### Reporting summary

Further information on research design is available in the Nature Portfolio Reporting Summary linked to this article.

## Data availability

Electron density maps have been deposited at the Electron Microscopy Database (accession codes EMDB-18471 and EMDB-18472), and atomic coordinates have been deposited at the PDB (PDB ID codes 8QKU and 8QKV). Initial models used for model building include PDB ID 6GEN and 6GEJ, as well as an AlphaFold-generated model of Swc2. Correspondence and requests for materials should be addressed to D.B.W. or D.S.R. All unique materials are available on request with completion of a standard Materials Transfer Agreement. Source data are provided with this paper.

## Code availability

Code for the single-molecule data analysis is freely available (https://github.com/singlemoleculegroup).

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

**Acknowledgements** We thank the Wigley and Rueda laboratories for comments and critical reading of the manuscript; Diamond for access and support of the cryo-EM facilities at the UK National Electron Bio-imaging Centre (eBIC), funded by the Wellcome Trust, Medical Research Council and BBSRC, and LonCEM facility; and N. Cronin for assistance with data collection at the LonCEM facility. The work was funded by the Wellcome Trust (095519/Z/11/Z and 209327/Z/17/Z (to D.B.W.)), Cancer Research UK (C6913/A21608 (to D.B.W.)), the Medical Research Council (MR/N009258/1 and MR/R009023/1 (to D.B.W.)), and a core grant from the MRC Laboratory of Medical Sciences (UKRIMC-A658-5TY10 (to D.S.R.)).

**Author contributions** P.G., A.S.B.J., D.S.R. and D.B.W. designed the studies. P.G. conducted and analysed the single-molecule experiments. A.S.B.J. performed the cryo-EM analysis. A.S.B.J., P.G., E.A.C., M.T.S. and C.L.K. prepared the samples. P.G., A.S.B.J., D.B.W. and D.S.R. analysed the data and wrote the manuscript with input from all the authors.

**Competing interests** The authors declare no competing interests.

**Additional information**
**Correspondence and requests for materials** should be addressed to Dale B. Wigley or David S. Rueda.

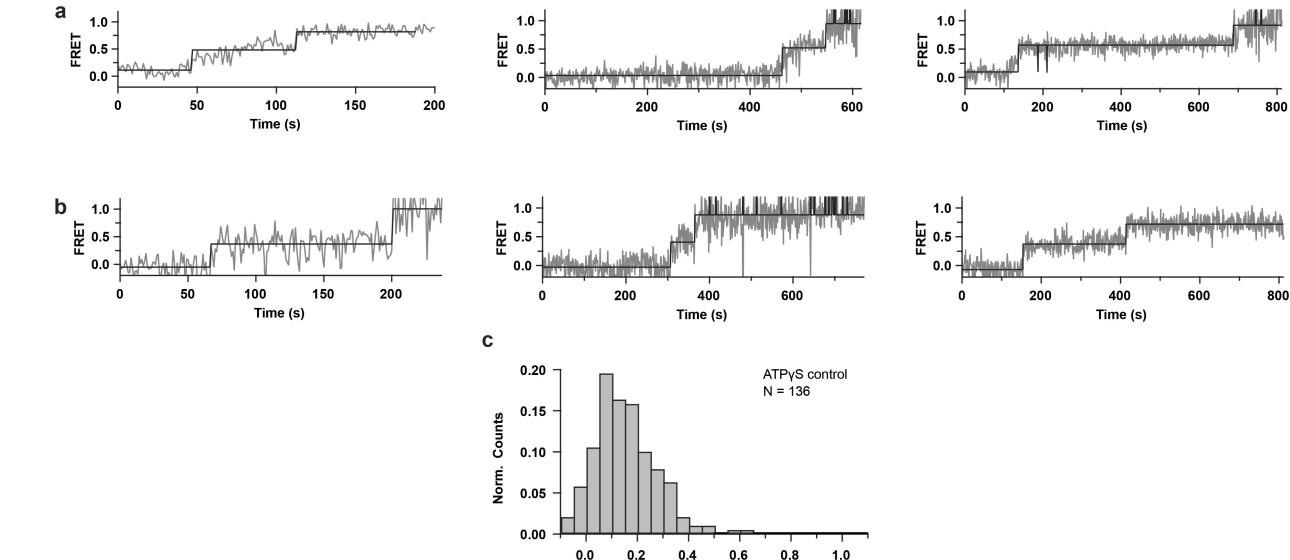

**Extended Data Fig. 1 | Additional examples of smFRET trajectories relating to the experiment described in Fig. 1 of the main text. a**, Three examples of double exchange events where the first exchange is on the dye proximal side. **b**, Three examples of double exchange events where the first exchange is on the dye distal side. **c**, Idealized FRET histogram of an exchange reaction carried out in the presence of ATPγS. No stepwise FRET increases like the examples shown in (**a**) and (**b**) are observed. Most molecules exhibit static low FRET.

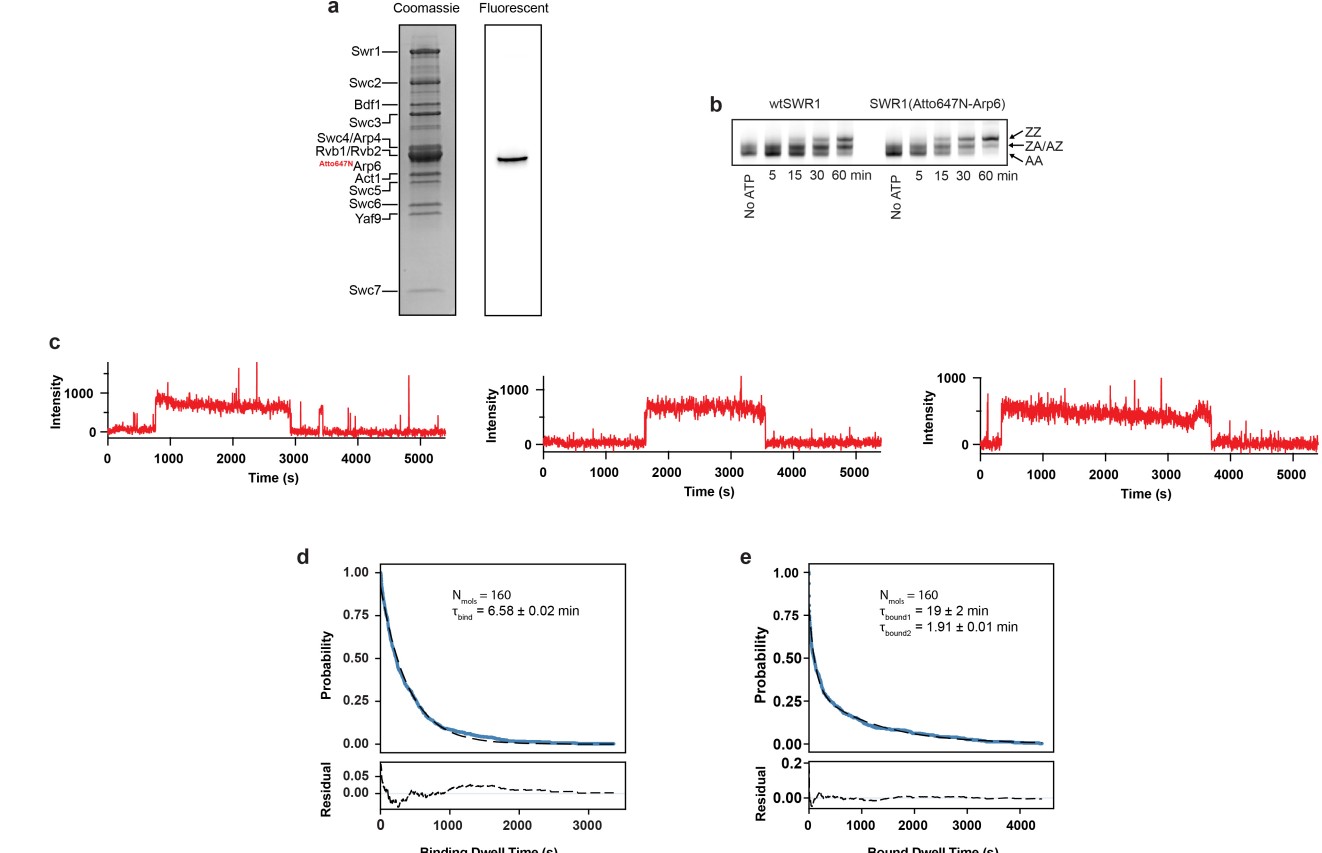

**Extended Data Fig. 2 | Fluorescently labelled SWR1 and measuring nucleosome bound lifetime. a**, SWR1 was specifically labelled with Atto647N on the N-terminus of the Arp6 subunit. Coomassie stained gel of the purified complex shows the presence of all expected SWR1 subunits (left). The same gel imaged for fluorescence shows that only the Arp6 subunit has been fluorescently labelled (right). Representative gel of three independent preparations. For gel source data, see Supplementary Fig. 1. **b**, Bulk activity assay using the insertion of a FLAG tagged Htz1–H2B dimer as a readout for exchange. Exchange activity of the labelled SWR1 complex is retained. Representative gel of two independent experiments using enzyme from separate purifications. For gel source data, see Supplementary Fig. 1. **c**, Three example single molecule intensity trajectories

of SWR1(647N) colocalization to surface immobilized nucleosomes. **d**, Dwell time plot of SWR1(647N) binding times. Data is shown fit to a single exponential decay (with residuals below). On average SWR1 takes $6.58 \pm 0.02$ min to bind under our experimental conditions. **e**, Dwell time plot of the lifetime of SWR1(647N) bound to a nucleosome. Data is shown fit to a double exponential decay (with residuals below). Two types of bound complex are present, one stably bound (lifetime $19 \pm 2$ min) and one more transiently bound (lifetime $1.91 \pm 0.01$ min). We tentatively assign the transiently bound species to SWR1(647N) interacting with the extranucleosomal DNA, and the stably bound species to SWR1(647N) engaging properly with the nucleosome. Reported errors are the error of the fit.

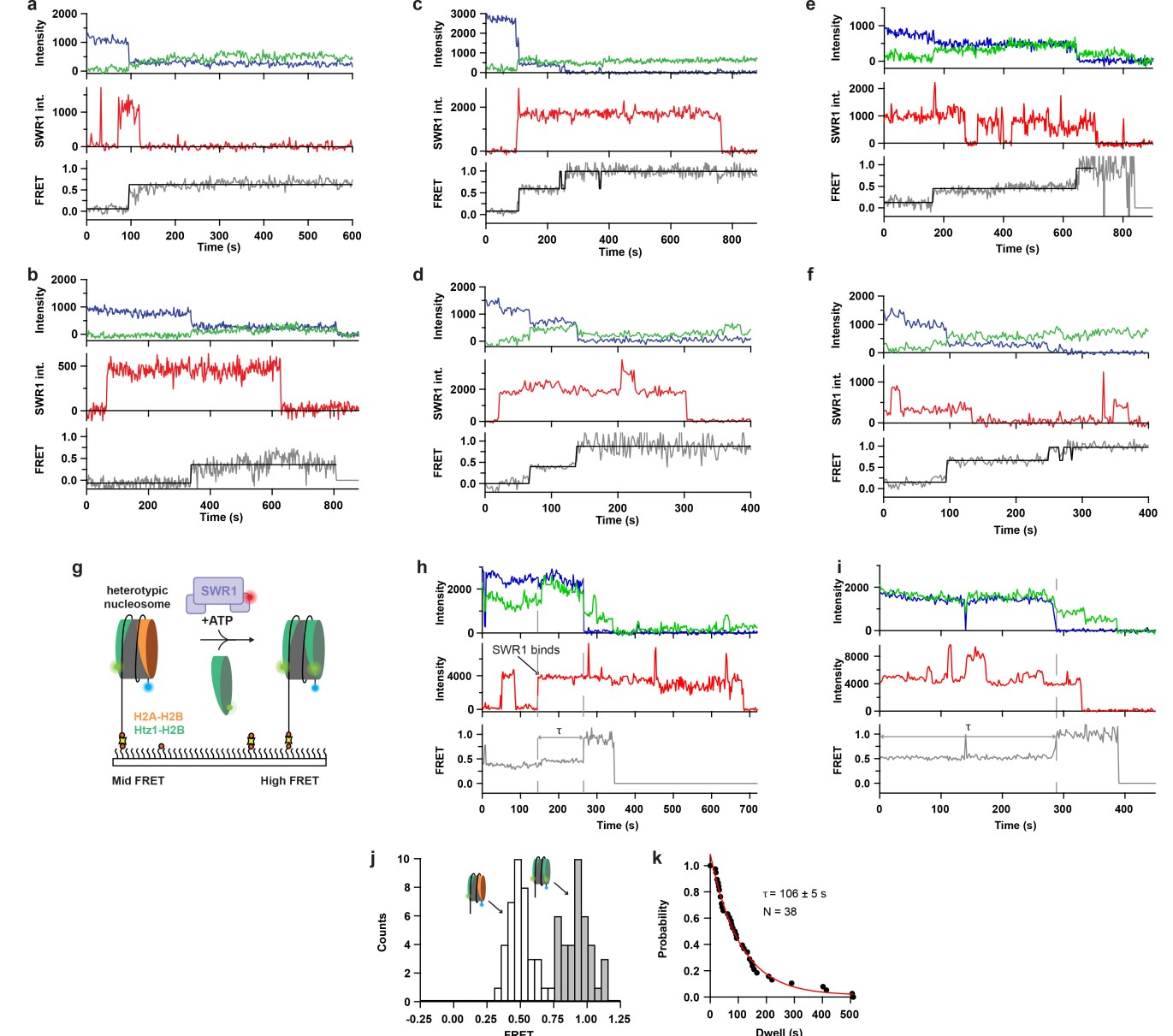

**Extended Data Fig. 3 | Additional examples of trajectories relating to the experiment described in Fig. 2 of the main text, and exchange of a heterotypic nucleosome. a**, & **b**, Single exchange events, where the exchange event is preceded by SWR1 binding. **c**, & **d**, Processive double exchange events. A single SWR1 binding event is followed by two consecutive exchange events. (Data in (**c**) is from Fig. 2d of the main text, replotted here to additionally show the donor and acceptor trajectories.) **e**, Distributive double exchange event. Following the first exchange SWR1 dissociates. The second exchange is preceded by a SWR1 binding event. **f**, Ambiguous double exchange example.

SWR1 either dissociates or photobleaches between the first and second exchange events. **g**, Schematic of three-color smFRET assay using a heterotypic nucleosome as the substrate. Schematic is colored similarly to Fig. 2a of the main text. **h**, & **i**, Example trajectories showing SWR1 binding and histone exchange of a heterotypic nucleosome substrate. **j**, Histogram showing the FRET before (white bars) and after (grey bars) exchange, for a heterotypic nucleosome. **k**, Distribution of the time between SWR1 binding and histone exchange yields an exchange time of 106 ± 5 s for a heterotypic nucleosome. Reported errors are the error of the fit.

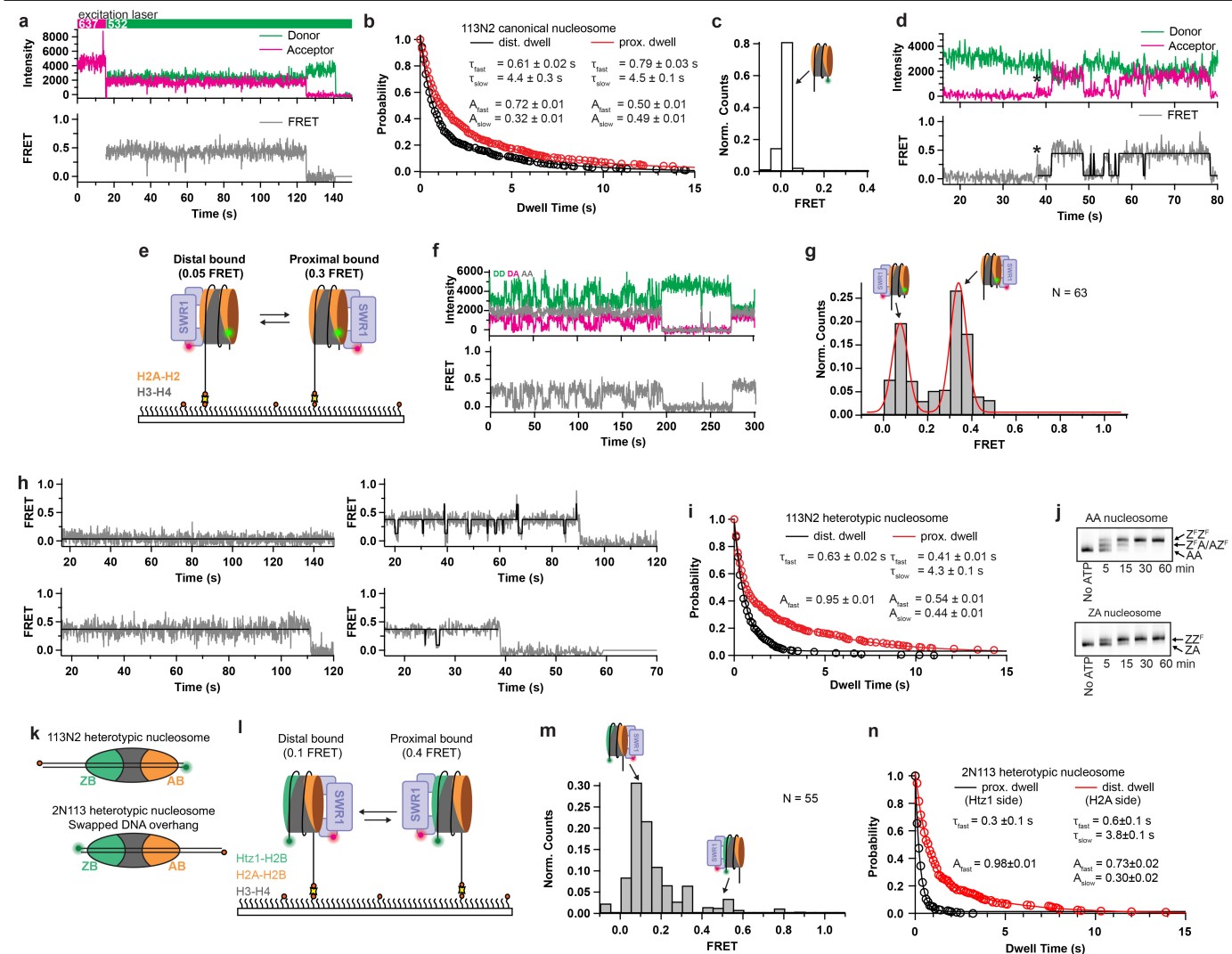

**Extended Data Fig. 4** | See next page for caption.

**Extended Data Fig. 4 | Additional data and controls relating to the experiments in Figs. 3 and 4 of the main text. a**, Example fluorescence intensity trajectory (top) and corresponding FRET trajectory (bottom) resulting from SWR1(647N) bound to a surface immobilized nucleosome. The excitation scheme used is illustrated with the magenta and green bars (top). After locating SWR1(647N) bound nucleosomes with red excitation, FRET between the nucleosome and SWR1(647N) is monitored using green excitation. Single step photobleaching of the acceptor and donor indicate a single FRET pair. **b**, Dwell time plot of the dwell times in the proximal or distal bound configurations for a nucleosome containing two H2A–H2B dimers (data from Fig. 3d, replotted here to show additional details of the fit). Dwell time is fit to a double exponential decay. The lifetimes in the fast ($\tau_{fast}$) and slow ($\tau_{slow}$) phases are indicated, along with the corresponding amplitudes ($A_{fast}$, $A_{slow}$). The lifetimes are approximately equal regardless of SWR1 orientation. **c**, FRET histogram of nucleosome (donor) only control displaying zero FRET in the absence of any SWR1(647N) (acceptor). **d**, Example fluorescence intensity trajectory (top) and corresponding FRET trajectory (bottom) showing SWR1(647N) binding to a surface immobilized nucleosome (indicated by *), and subsequently flipping between dye-distal and dye-proximal orientations. SWR1(647N) binding results in a small but detectable non-zero FRET. (Note: such a trajectory would not be included in subsequent analysis as it does not satisfy the criterion of having SWR1(647N) bound at the start of data acquisition, but is shown here to illustrate detection of SWR1(647N) binding and flipping.) **e**, Schematic of the assay where the donor fluorophore is placed on one of the H2A histones: Nucleosomes (113N2) labelled with Cy3B on the linker-distal H2A are surface immobilized. SWR1(647N) is flowed in and allowed to bind the nucleosomes. SWR1(647N)–nucleosome interactions are monitored via FRET. Repositioning the FRET donor from the short DNA overhang (as used throughout the rest of this work) to the linker-distal H2A results in lower FRET efficiencies. To identify these true low-FRET values we employed alternating laser excitation throughout the entire acquisition. **f**, Trajectory of a dynamic SWR1(647N) bound nucleosome showing donor emission upon donor excitation (DD, green trace, top); acceptor emission upon donor excitation (DA, magenta trace, top); acceptor emission upon acceptor excitation (AA, gray trace, top). DD and DA are used for calculating apparent FRET efficiency (gray trace, bottom). **g**, Idealized FRET histogram shows two major populations of SWR1(647N)–bound nucleosomes. We observe similar ratios of the two states regardless of FRET donor position (c.f. Fig. 3). **h**, Additional smFRET traces from the experiment described in Fig. 4b. **i**, Dwell time plot of the dwell times in the proximal or distal bound configurations for a heterotypic nucleosome containing one Htz1–H2B and one H2A–H2B dimer (data from Fig. 4d, replotted here to show additional details of the fit). While the time spent on the distal side (i.e., the side containing Htz1) is well described by a single exponential decay, the proximal (H2A containing side) is best fit to a double exponential decay. Compare with (**b**). The lifetimes in the fast ($\tau_{fast}$) and slow ($\tau_{slow}$) phases are indicated, along with the corresponding amplitudes ($A_{fast}$, $A_{slow}$). **j**, Bulk assay showing the exchange of a canonical H2A–H2B nucleosome (AA) compared to a heterotypic nucleosome containing one Htz1–H2B and one H2A–H2B dimer (ZA). The insertion of a FLAG tagged Htz1–H2B dimer is used as a readout for exchange. The AA nucleosome undergoes two consecutive rounds of exchange (indicated by the appearance of a double band shift). However, the ZA nucleosome can only be exchanged once (single band shift) indicating that SWR1 does not remove Htz1–H2B dimers from a nucleosome. Representative gel of two independent experiments using enzyme from separate purifications. For gel source data, see Supplementary Fig. 1. **k**, Cartoons of 113N2 heterotypic nucleosome (top) and 2N113 swapped DNA overhang heterotypic nucleosome (bottom). The position of the Cy3 fluorophore (green circle) and biotin (orange circle) are shown. Swapping the DNA overhang orientation with respect to the 601 positioning sequence results in the Htz1–H2B variant histone either being adjacent or opposite to the long DNA overhang. **l**, Swapped DNA overhang heterotypic nucleosomes (Cy3.2N113) containing one Htz1–H2B dimer (green) and one canonical H2A–H2B dimer (orange) Cy3-labeled on the 2 bp overhang are surface immobilized. SWR1[647N] is flowed in and allowed to bind to the nucleosome. SWR1(647N)–nucleosome interactions are monitored via FRET. **m**, Idealized FRET histogram shows a main population at low (0.1) FRET (c.f. Fig. 4c). **n**, Dwell time plots for the distal to proximal (red) and proximal to distal (black) transition for a 2N113 heterotypic nucleosome. Binding to the H2A–H2B face of the nucleosome is more stable than binding to the Htz1–H2B face, irrespective of the location of the long DNA overhang.

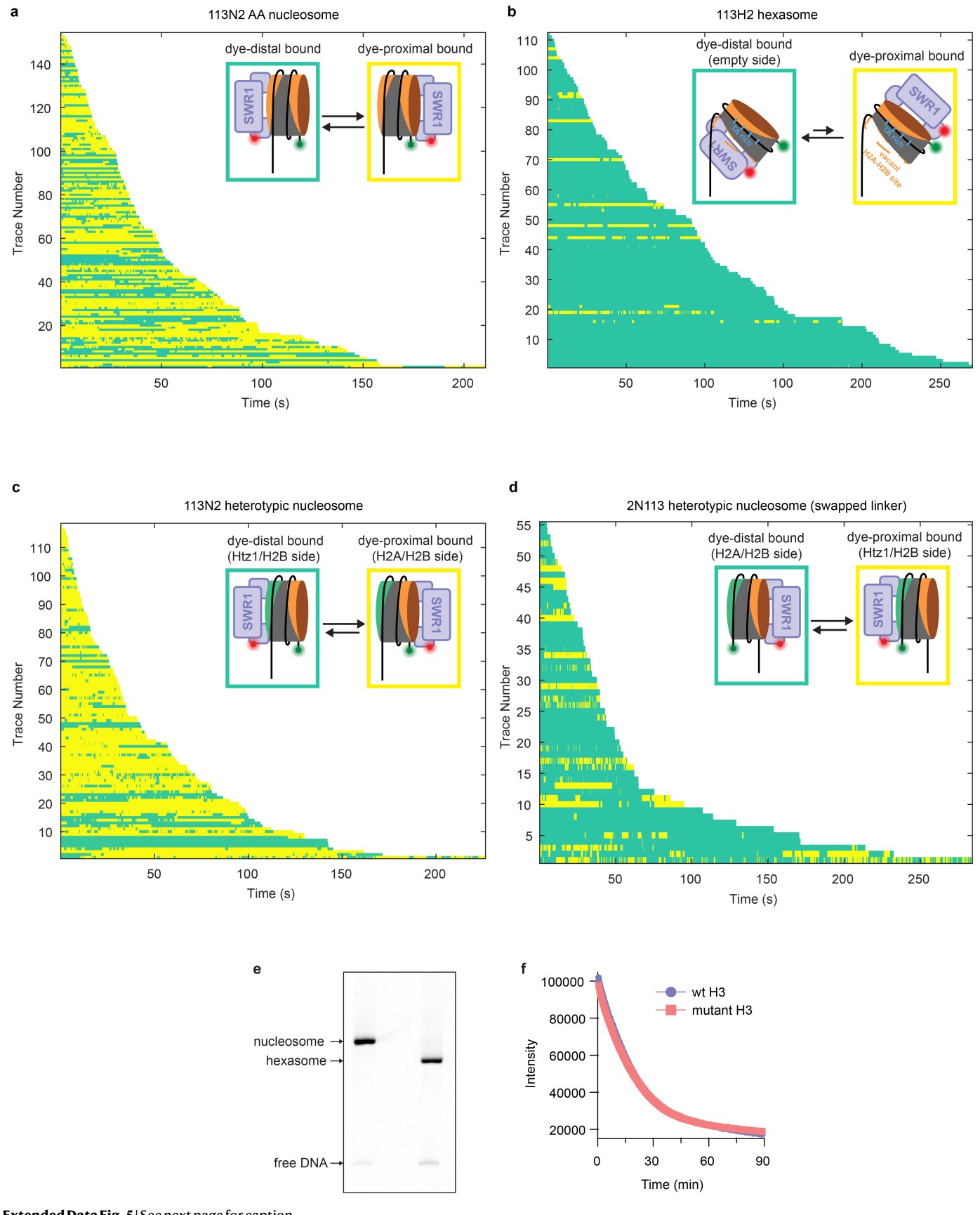

**Extended Data Fig. 5** | See next page for caption.

**Extended Data Fig. 5 | Rastergrams summarising the nucleosome flipping data for different nucleosomes/hexasome, and preparation of yeast hexasomes. a-d**, Each horizontal line represents a smFRET trajectory, ordered by photobleaching/dissociation time. Color indicates whether SWR1 is bound in the dye-distal (green) or dye-proximal (yellow) orientations. Thresholding (at 0.25 FRET) of the idealized FRET trajectories was used to determine the two states. Data is shown for: **a**, Canonical H2A–H2B 113N2 nucleosomes. **b**, Hexasomes 113H2 containing only one H2A–H2B dimer. **c**, Heterotypic nucleosomes 113N2 containing both H2A–H2B and Htz1–H2B histones. **d**, Swapped linker heterotypic nucleosomes 2N113 containing both H2A–H2B and Htz1–H2B histones. **e**, Native PAGE comparing a nucleosome and hexasome sample. Representative gel of two independent preparations. **f**, H3MPQ mutations required for formation of *S. cerevisiae* hexasomes and heterotypic nucleosomes (see Methods) have no effect on SWR1 exchange activity as measured by bulk FRET decrease.

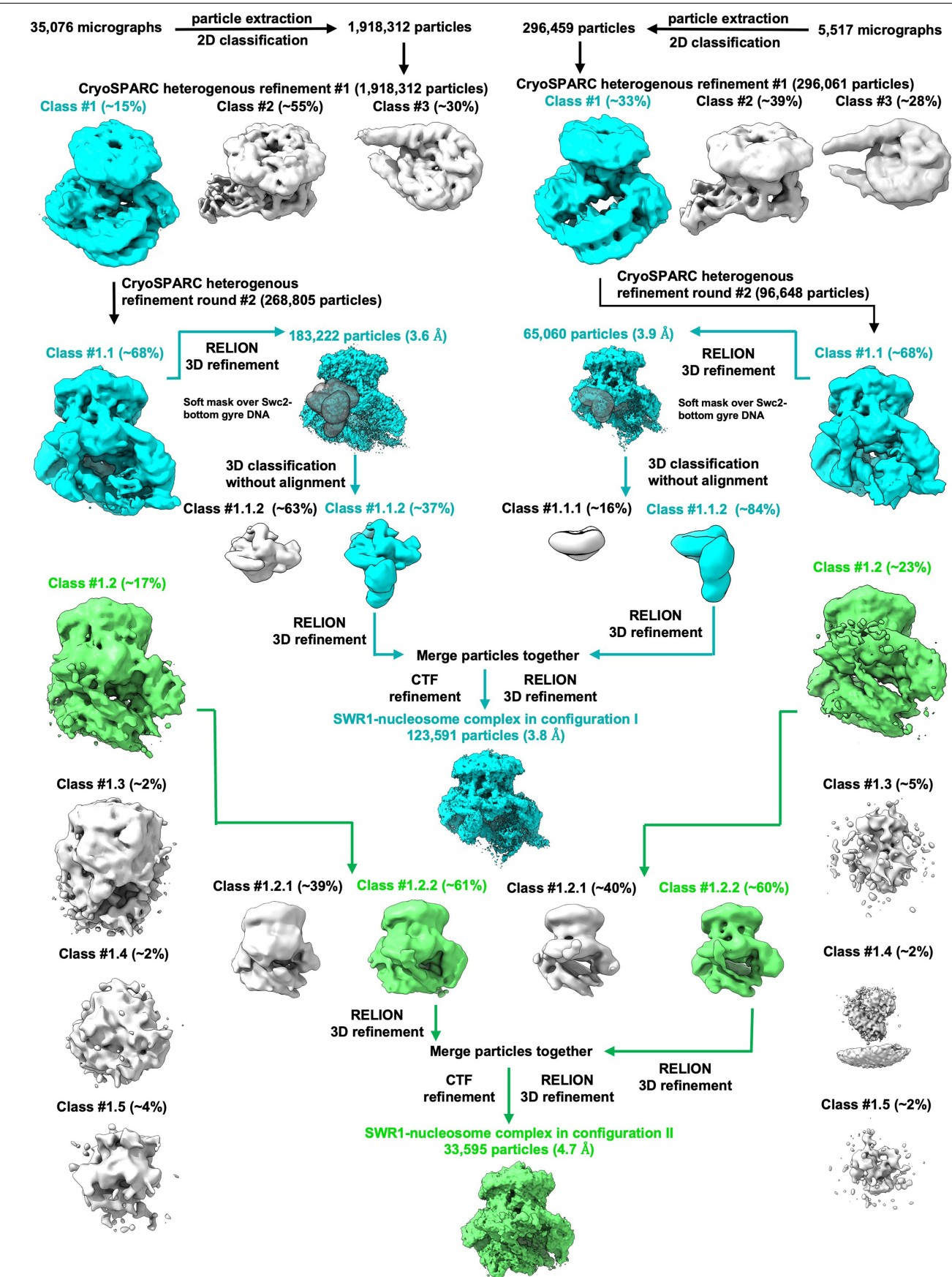

**Extended Data Fig. 6 | Schematic overview of the cryoEM processing.** Schemes for the 3.8 Å SWR1–nucleosome complex in configuration I (cyan) and the 4.7 Å SWR1–nucleosome complex in configuration II (green) datasets.

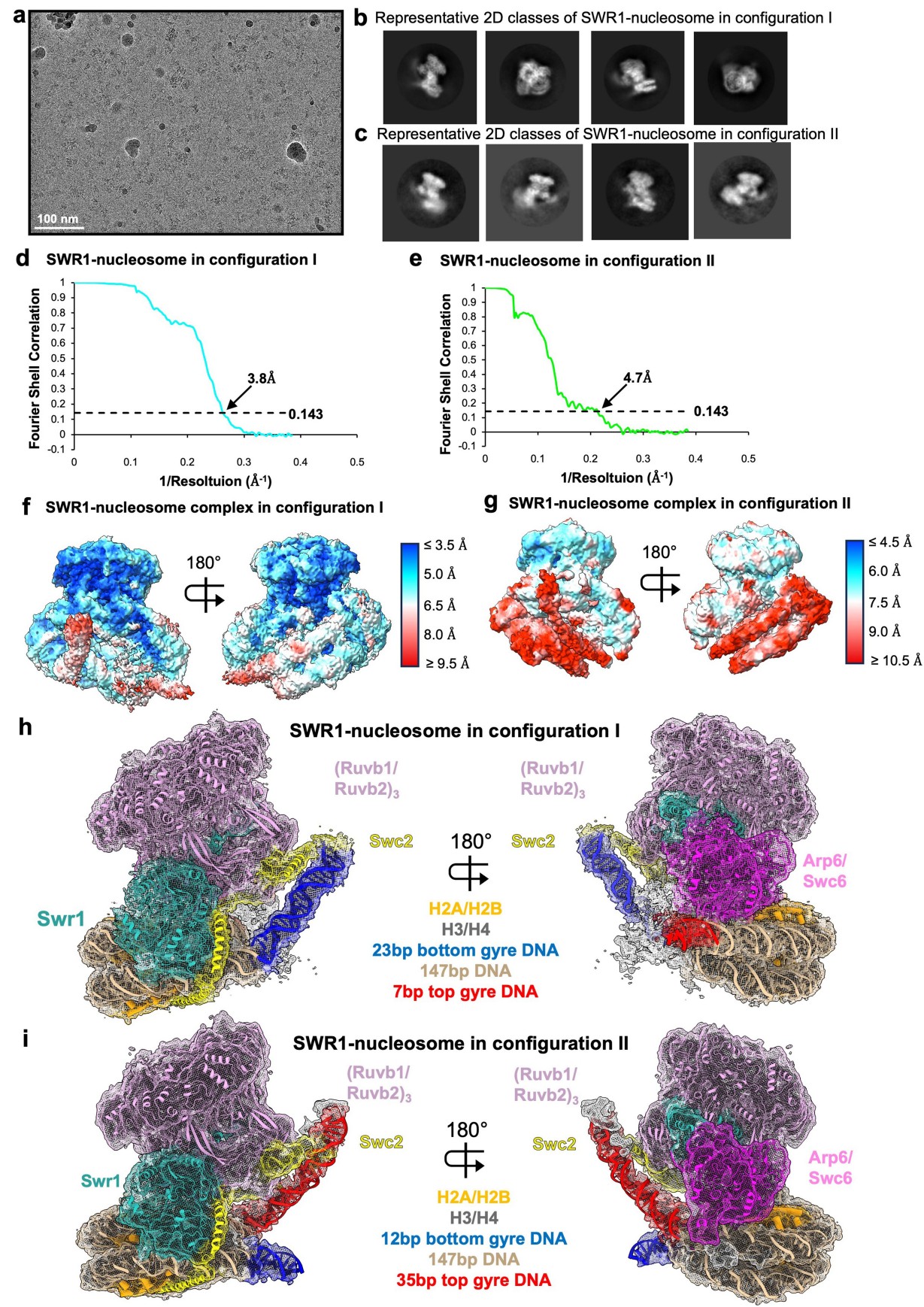

**Extended Data Fig. 7** | See next page for caption.

**Extended Data Fig. 7 | Cryo-EM analysis of the SWR1–nucleosome in configuration I (3.8 Å) and SWR1–nucleosome in configuration II (4.7 Å) volumes. a**, Representative micrograph out of 35,076 micrographs from the SWR1–nucleosome dataset. A scale bar is shown at the bottom left. **b**, Four representative 2D classes of SWR1–nucleosome complex in configuration I. **c**, Four representative 2D classes of SWR1–nucleosome complex in configuration II. **d**, gFSC curve of the SWR1–nucleosome in configuration I volume. **e**, gFSC curve of the SWR1–nucleosome configuration II volume. **f**, Local resolution of the SWR1–nucleosome complex in configuration I. **g**, Local resolution of SWR1–nucleosome in configuration II. **h**, Overview of the SWR1–nucleosome complex in configuration I at 3.8 Å. **i**, Overview of the SWR1–nucleosome complex in configuration II at 4.7 Å.

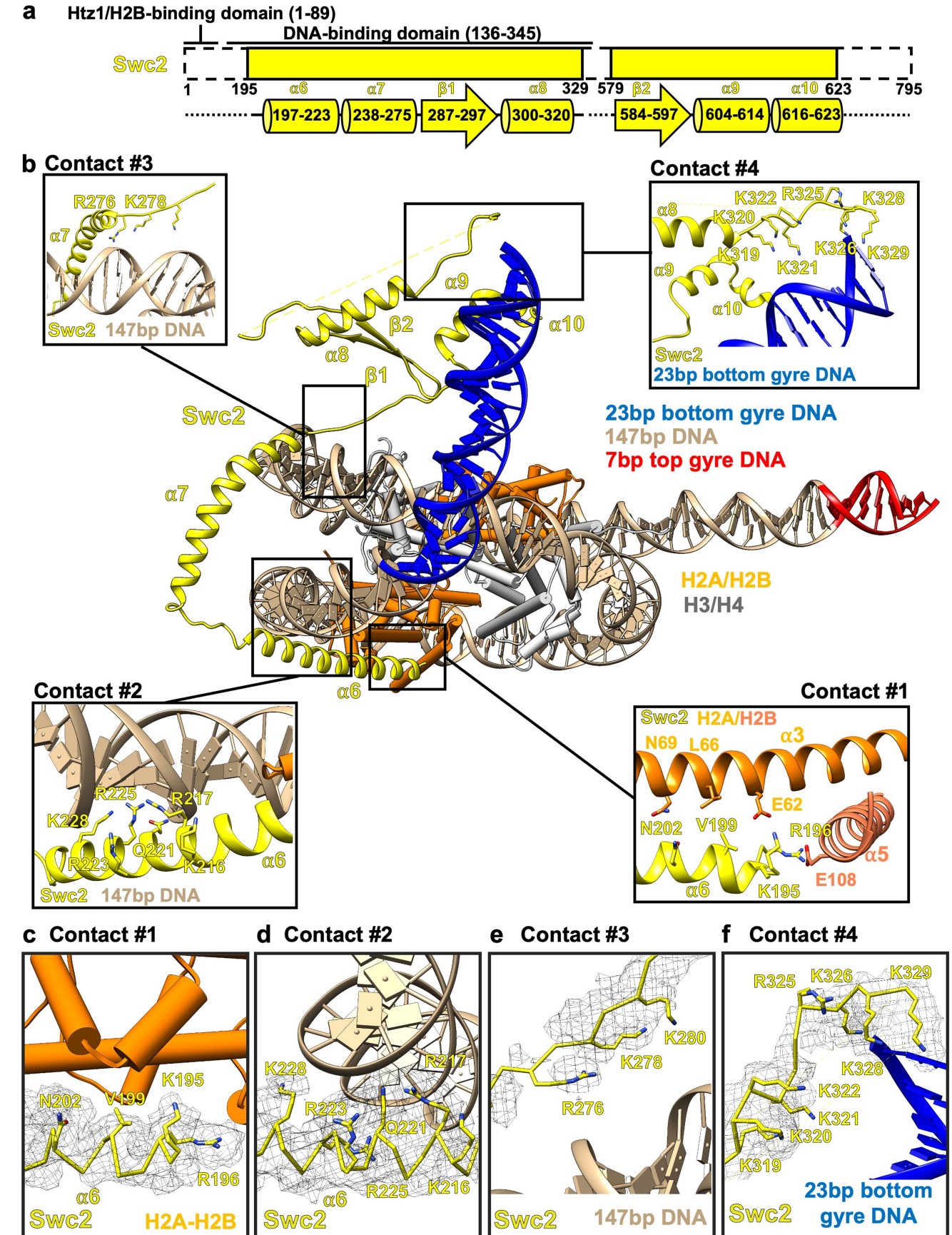

**a** **Htz1/H2B-binding domain (1-89)**

Swc2

**DNA-binding domain (136-345)**

1  195  α6  α7  β1  α8  329  579  β2  α9  α10  623  795

197-223  238-275  287-297  300-320  584-597  604-614  616-623

**b** Contact #3

R276  K278
α7
Swc2  147bp DNA

Contact #4

K322  R325  K328
K320
α8  K319  K326  K329
α9  K321
α10
Swc2  23bp bottom gyre DNA

Swc2

α9  α8  β2
β1  α10

Swc2

α7

23bp bottom gyre DNA
147bp DNA
7bp top gyre DNA

H2A/H2B
H3/H4

α6

Contact #2

R225  R217
K228  Q221
R223  K216  α6
Swc2  147bp DNA

Contact #1

Swc2  H2A/H2B
N69  L66  α3
E62
N202  V199  R196
α6  K195  E108  α5

**c** Contact #1

N202  V199  K195
α6  R196
Swc2  H2A-H2B

**d** Contact #2

K228  R217
R223  Q221
R225  K216
Swc2  α6

**e** Contact #3

K280
K278
R276
Swc2  147bp DNA

**f** Contact #4

R325  K326  K329
K328
K322
K321
K320
K319
Swc2  23bp bottom gyre DNA

**Extended Data Fig. 8** | See next page for caption.

**Extended Data Fig. 8 | Details of the interaction between Swc2 and the nucleosome in the SWR1-nucleosome in configuration I. a**, Linearized cartoon of the Swc2 subunit, the built-in coordinates are represented in yellow. The Htz1–H2B binding domain (residues 1–89) and the DNA-binding domain (residues 136–345) are highlighted. **b**, The residues of Swc2 that binds the DNA or interact with the H2A–H2B histones are highlighted The interaction between Swc2 and the nucleosome in the SWR1–nucleosome complex in configuration I. For simplicity, only the built in coordinates of Swc2 and the nucleosome is shown. **c**, Representative density for Swc2 at contact 1 (contoured at 2σ) in the SWR1–nucleosome in configuration I complex. **d**, Representative density for Swc2 at contact 2 (contoured at 2.5σ) in the SWR1–nucleosome in configuration I complex. The side chains of Swc2 that interacts with nucleosomal DNA is shown. **e**, Representative density for Swc2 at contact #3 (contoured at 5.5σ). in the SWR1–nucleosome in configuration I complex. The side chains of Swc2 that interacts with the nucleosomal DNA is shown. **f**, Representative density for Swc2 at contact #4 (contoured at 3σ). in the SWR1–nucleosome in configuration I complex. The side chains of Swc2 that interacts with the bottom gyre nucleosomal DNA is shown.

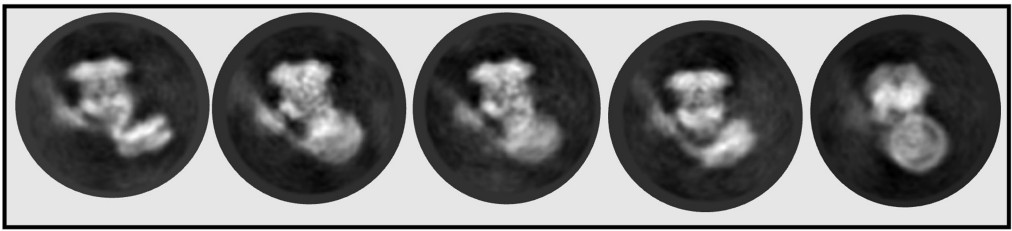

**a**

**Contact #3**

4
3
bits 2
1
0

L Q N R K L K L
M N Q S Q K F
+        +        +        +

271                    281

**+ DNA-biding residues**

**Contact #4**

3
bits 2
1
0

K K K R R K K K
R R R Y G P K R S N
E T T L T S P D G
P E E P R S T Q A G
+   +        +   +   +   +

319                    329

**+ DNA-biding residues**

Swc2

α7

α8  β2

β1

α9

α10

23bp bottom gyre DNA

147bp DNA

7bp top gyre DNA

H2A/H2B

H3/H4

α6

**Contact #2**

4
3
bits 2
1
0

E R R K L A T M E K A A
K K K A K T V E R I K E K
K K V L K O P L R H V
+        +        +        +        +

215                    228

**+ DNA-biding residues**

**Contact #1**

4
3
bits 2
1
0

S R S S T V N
K K R Q A M S
E T Q M G S
*        *        *        *

195                    202

**\* H2A-H2B binding residues**

**b** **Additional 2D classes of SWR1-mediated nucleosome flipping**

**Extended Data Fig. 9 | Residues that interact with the nucleosome are conserved between Swc2-like proteins, and additional 2D classes of SWR1-mediated nucleosome flipping. a**, The interaction between Swc2 and the nucleosome in the SWR1–nucleosome complex in configuration I. For simplicity, only the built-in coordinates of Swc2 and the nucleosome are shown. The four different contacts between Swc2 and the nucleosome are highlighted.

An alignment of 116 Swc2-like proteins across various species was used to generate a sequence logo to display sequence conservation. The residues of Swc2 that bind the DNA or interact with the H2A–H2B histones are highlighted. **b**, Five additional intermediates of SWR1-mediated nucleosome flipping are visible after 2D classification.

**Extended Data Table 1 | Cryo-EM data collection, refinement and validation statistics**

| | SWR1-nucleosome complex in configuration I (EMDB-18471) (PDB 8QKU) | SWR1-nucleosome complex in configuration II (EMDB-18472) (PDB 8QKV) |
|---|---|---|
| **Data collection and processing** | | |
| Magnification | 80,000 | 80,000 |
| Voltage (kV) | 300 | 300 |
| Electron exposure (e–/Å²) | 50 | 50 |
| Defocus range (μm) | -0.7 to -1.9 | -0.7 to -1.9 |
| Pixel size (Å) | 1.1 | 1.1 |
| Symmetry imposed | C1 | C1 |
| Initial particle images (no.) | 365,453 | 365,453 |
| Final particle images (no.) | 123,591 | 33,595 |
| Map resolution (Å) | 3.8 | 4.7 |
| FSC threshold | 0.143 | 0.143 |
| Map resolution range (Å) | 3.5 to >9.5 | 4.5 to >10.5 |
| | | |
| **Refinement** | | |
| Initial model used (PDB code) | 6GEJ, 6GEN, AlphaFold | 6GEJ, 6GEN, AlphaFold |
| Model resolution (Å) | 3.8 | 4.7 |
| FSC threshold | 0.143 | 0.143 |
| Model resolution range (Å) | 3.5 to >9.5 | 4.5 to >10.5 |
| Map sharpening $B$ factor (Å²) | -90 | -80 |
| Model composition | | |
| Non-hydrogen atoms | 45395 | 45846 |
| Protein residues | 4863 | 4840 |
| Nucleotide | 354 | 338 |
| Ligands | 20 | 20 |
| $B$ factors (Å²) | | |
| Protein (mean) | 155.71 | 759.39 |
| Nucleotide (mean) | 314.96 | 1100.1 |
| Ligand (mean) | 131.22 | 644.28 |
| R.m.s. deviations | | |
| Bond lengths (Å) | 0.003 | 0.003 |
| Bond angles (°) | 0.640 | 0.638 |
| Validation | | |
| MolProbity score | 1.92 | 2.09 |
| Clashscore | 13.12 | 18.06 |
| Poor rotamers (%) | 0.29 | 0.02 |
| Ramachandran plot | | |
| Favored (%) | 95.80 | 95.13 |
| Allowed (%) | 4.14 | 4.76 |
| Disallowed (%) | 0.06 | 0.10 |

# Reporting Summary

## Statistics

For all statistical analyses, confirm that the following items are present in the figure legend, table legend, main text, or Methods section.

| n/a | Confirmed | |
|---|---|---|
| ☐ | ☒ | The exact sample size (*n*) for each experimental group/condition, given as a discrete number and unit of measurement |
| ☐ | ☒ | A statement on whether measurements were taken from distinct samples or whether the same sample was measured repeatedly |
| ☒ | ☐ | The statistical test(s) used AND whether they are one- or two-sided *Only common tests should be described solely by name; describe more complex techniques in the Methods section.* |
| ☒ | ☐ | A description of all covariates tested |
| ☒ | ☐ | A description of any assumptions or corrections, such as tests of normality and adjustment for multiple comparisons |
| ☐ | ☒ | A full description of the statistical parameters including central tendency (e.g. means) or other basic estimates (e.g. regression coefficient) AND variation (e.g. standard deviation) or associated estimates of uncertainty (e.g. confidence intervals) |
| ☒ | ☐ | For null hypothesis testing, the test statistic (e.g. *F*, *t*, *r*) with confidence intervals, effect sizes, degrees of freedom and *P* value noted *Give P values as exact values whenever suitable.* |
| ☐ | ☒ | For Bayesian analysis, information on the choice of priors and Markov chain Monte Carlo settings |
| ☒ | ☐ | For hierarchical and complex designs, identification of the appropriate level for tests and full reporting of outcomes |
| ☒ | ☐ | Estimates of effect sizes (e.g. Cohen's *d*, Pearson's *r*), indicating how they were calculated |

*Our web collection on statistics for biologists contains articles on many of the points above.*

## Software and code

Policy information about availability of computer code

| Data collection | Single-molecule videos were acquired using a home-built LabView scripts or HCImage 4.6.1.3 (Hammamatsu). CryoEm data was collected on a Titan Krios microscope. |
|---|---|
| Data analysis | Single-molecule data extraction: IDL 8.4. Single-molecule data viewing: MATLAB R2022b. Single-molecule HMM analysis: vbFRET or tMAVEN 0.2.0. Single-molecule data plotting: Igor Pro 8.04. All custom analysis scripts used can be found on the groups github page: (https://github.com/singlemoleculegroup). CryoEM data processing: MotionCor2, CTFFIND4, cryoSPARC 3.3.2, RELION 4.0, PHENIX 1.20.1, UCSF Chimera 1.16, COOT 0.9.8.3 , ChimeraX 1.6.1, AlphaFold 2.1.0 |

For manuscripts utilizing custom algorithms or software that are central to the research but not yet described in published literature, software must be made available to editors and reviewers. We strongly encourage code deposition in a community repository (e.g. GitHub). See the Nature Portfolio guidelines for submitting code & software for further information.

## Data

Policy information about availability of data

All manuscripts must include a data availability statement. This statement should provide the following information, where applicable:

- Accession codes, unique identifiers, or web links for publicly available datasets
- A description of any restrictions on data availability
- For clinical datasets or third party data, please ensure that the statement adheres to our policy

Electron density maps are deposited at the Electron Microscopy Database (accession codes EMDB-18471 & EMDB-18472) and atomic coordinates are deposited at the Protein Databank (PDB ID codes 8QKU & 8QKV). Initial models used for model building include PDB ID:6GEN & 6GEJ, as well as an AlphaFold generated model of Swc2. The datasets generated during and/or analysed during the current study will be available from the corresponding author on reasonable request.

## Research involving human participants, their data, or biological material

Policy information about studies with human participants or human data. See also policy information about sex, gender (identity/presentation), and sexual orientation and race, ethnicity and racism.

| | |
|---|---|
| Reporting on sex and gender | N/A |
| Reporting on race, ethnicity, or other socially relevant groupings | N/A |
| Population characteristics | N/A |
| Recruitment | N/A |
| Ethics oversight | N/A |

Note that full information on the approval of the study protocol must also be provided in the manuscript.

# Field-specific reporting

Please select the one below that is the best fit for your research. If you are not sure, read the appropriate sections before making your selection.

☒ Life sciences  ☐ Behavioural & social sciences  ☐ Ecological, evolutionary & environmental sciences

For a reference copy of the document with all sections, see nature.com/documents/nr-reporting-summary-flat.pdf

# Life sciences study design

All studies must disclose on these points even when the disclosure is negative.

| | |
|---|---|
| Sample size | No prior sample size calculation was performed. Sample sizes were selected based on previous experience. All observations were made on sufficient numbers of individual molecules, when possible more than 100. For structural determination, the number of micrographs in our cryoEM data collection was chosen accordingly to obtain the required resolution. |
| Data exclusions | Inclusion criteria for single-molecule data are detailed in the Methods section. |
| Replication | Single-molecule data was independently replicated at least twice. The total number of traces used for each dataset are indicated on each figure. For bulk assays (gels) two independant repeats were performed one of which is shown, attempts at replication were successful. |
| Randomization | In the Fourier shell correlation (FSC) measurement in RELION 4.0 pipeline, data from the Refine3D job was randomly divided into two halves resulting in two independently determined 3D volumes that were used for the FSC calculation through a Postprocess job. |
| Blinding | Blinding was not relevant to the experiments in this study. Cryo-EM and biochemcial data were collected and processed in an unbiased manner. |

# Reporting for specific materials, systems and methods

We require information from authors about some types of materials, experimental systems and methods used in many studies. Here, indicate whether each material, system or method listed is relevant to your study. If you are not sure if a list item applies to your research, read the appropriate section before selecting a response.

## Materials & experimental systems

| n/a | Involved in the study |
|-----|----------------------|
| ☒ | Antibodies |
| ☐ | ☒ Eukaryotic cell lines |
| ☒ | Palaeontology and archaeology |
| ☒ | Animals and other organisms |
| ☒ | Clinical data |
| ☒ | Dual use research of concern |
| ☒ | Plants |

## Methods

| n/a | Involved in the study |
|-----|----------------------|
| ☒ | ChIP-seq |
| ☒ | Flow cytometry |
| ☒ | MRI-based neuroimaging |

## Eukaryotic cell lines

Policy information about cell lines and Sex and Gender in Research

| | |
|---|---|
| Cell line source(s) | Spodoptera frugiperda Sf9, Thermo Fisher Scientific, 11496015<br>Trichoplusia ni High Five,Thermo Fisher Scientific, B85502 |
| Authentication | Cell lines not authenticated |
| Mycoplasma contamination | Cell lines were not tested for mycoplasma |
| Commonly misidentified lines (See ICLAC register) | None used |

## Plants

| | |
|---|---|
| Seed stocks | *Report on the source of all seed stocks or other plant material used. If applicable, state the seed stock centre and catalogue number. If plant specimens were collected from the field, describe the collection location, date and sampling procedures.* |
| Novel plant genotypes | *Describe the methods by which all novel plant genotypes were produced. This includes those generated by transgenic approaches, gene editing, chemical/radiation-based mutagenesis and hybridization. For transgenic lines, describe the transformation method, the number of independent lines analyzed and the generation upon which experiments were performed. For gene-edited lines, describe the editor used, the endogenous sequence targeted for editing, the targeting guide RNA sequence (if applicable) and how the editor was applied.* |
| Authentication | *Describe any authentication procedures for each seed stock used or novel genotype generated. Describe any experiments used to assess the effect of a mutation and, where applicable, how potential secondary effects (e.g. second site T-DNA insertions, mosiacism, off-target gene editing) were examined.* |

