## [Peer Review file · Nature]

Manuscript Title: Nucleosome flipping drives kinetic proofreading and processivity by SWR1

Redactions – Third Party Material

Reviewer Comments & Author Rebuttals

Reviewer Reports on the Initial Version:

Referees' comments:

Referee #1 (Remarks to the Author):

The paper from Girvan et al. is highly insightful and delves into the mechanism of H2A.Z exchange for H2A by the SWR1 chromatin remodeler using a variety of single molecule FRET based assays coupled with some cryoEM analysis. They provide evidence for SWR1 working processively such that after one dimer is replaced the enzyme remains bound and then subsequently exchanges the second dimer. The three-color FRET assay is particularly informative that allows dimer exchange to be tracked at the same time while following binding of SWR1 to nucleosomes.

The FRET assay with labeled nucleosome and SWR1 to detect flipping of the nucleosome when associated with SWR1 is well designed. However, I had a question as to how well the distal bound complex with SWR1 bound to the other side of the nucleosome away from the fluorescent donor is distinguished from background noise. It seems from the traces that the 0.1 FRET signal is so close to the noise level that it might not be that easy to distinguish the two from each other. Although the simplest interpretation of these data is the orientation of the nucleosome is flipped could the authors also explain other potential interpretations of the data and why these would be less favored? Could the authors explain why there is the discrepancy between their and others' reports as to whether the dimer proximal to the linker DNA is preferred to be exchanged or not. Is this the difference between yeast versus human/Xenopus hybrid nucleosomes and if so why is that?

In the heterotypic nucleosome FRET assays, could the linker DNA play a contributing role for SWR1 preferring to reside longer on the proximal side where the H2A/H2B resides? It would be important to reverse the nucleosome orientation such that the H2A/H2B dimer is on the long linker side of nucleosomes to confirm that linker DNA is not a contributing factor. I worry the linker DNA could be a factor based on how SWR1 interacts with the longer linker DNA.

In the abstract the authors state that "CryoEM analysis reveals different populations of complexes showing how nucleosomes 'flip' between different conformation without release". I think this is a bit of an overstatement because as discussed in the main text due to the dynamic flipping there are only to visualize well a small number of conformations that are likely in the flipping state and are not sufficient to show how nucleosomes are flipped only that they have the potential to be.

Referee #2 (Remarks to the Author):

The exchange of the canonical H2A with the variant H2A.Z, Htz1 in yeast, alter the chromatin structure, and plays an important role in gene regulation. This manuscript by Girvan and colleagues explore the mechanism of Htz1-H2B dimer exchange by the yeast SWR1 complex using smFRET and cryoEM. The smFRET data showed that: 1. the reaction of double dimer exchange can be processive without the release of SWR1 from its substrate nucleosome; 2. SWR1 flips between the two faces of a nucleosome, which was regulated by the histone compositions. Additionally, they solved the cryo-EM structure of SWR1 bound to a nucleosome with the extended linker DNA bound by the Swc2 subunit.

The kinetics of the H2A.Z exchange reaction catalyzed by the SWR1 complex has been extensively studied, and the authors provide some new interesting findings. However, the current interpretations of the data at several places are not convincing.

Major points:

1. Using the 113N2 nucleosome as the substrate, the authors found the equal propensity for exchange of each dimer in a yeast nucleosome (Fig. 1c). This is at odd with the several earlier studies in vitro. The authors interpretate this discrepancy as the difference of the histones used. However, this interpretation is not convincing, as the failure to detect the linker DNA dependence of the exchange reaction may simply arise because of the other reagents used, including the DNA sequence, or other kinds of technical incompetence. Notably, the authors used recombinant SWR1 complex in this study, whereas the native complex was used by Poyton, Fan and Singh (ref. 18,19, and 20) in the previous studies. More importantly, the current finding is inconsistent with the asymmetric deposition of the Htz1 variant inside the cells.
2. The authors claimed that “we did not observe any distributive exchange”. But a distributive case was shown in Fig. S4G, and possibly another case in S4H. This raises a question how processive the enzyme can be, which is probably dependent on the experimental conditions, the salt concentration in particular.
3. In Fig.3, the authors claim that SWR1 flips between the two faces of a nucleosome. However, the altered FRET values can also be interpreted by the unwrapping or bending of the DNA. DNA bending and unwrapping are not the remote possibilities, considering the large conformational changes of the DNA bound by SWR1, as shown in Fig.5 and the previous study. I suggest the authors place the probes at more stable positions, such as labeling the H2A-H2B dimer, and repeat the assay. Or the authors can provide evidences to eliminate these possibilities.
4. To indicate a different binding state in Fig. 3b, a low FRET value of 0.1 is used, which is, however, barely above the background noise. Quantification of the low/high FRET ratio is probably error prone.
5. SWR1 is well known to selectively exchange the H2A-H2B for the Htz1-H2B. The findings in Fig. 4a-d provide little new insight into the selectivity, but is confirmatory to the previous studies. The stable binding to the hexasome shown in Fig, 4e-4f is very interesting. However, the authors offer little insights into this intermediate. As discussed above, a low FRET value of 0.1 for a prominent new state is less satisfactory. Placing the empty surface of the hexasome at the proximal site will provide much convincing evidence.
6. A longer section of overhang DNA is evident that emanates from the the nucleosome. However, it

is hard to tell by the current data shown in S11 A and B from which DNA gyres the EM density come. The maps are noisy, and the connectivity does not seem to support a clear identity of the DNA gyres.

7. To explain the flipping behavior and the processivity observed in FRET experiments, a model in Fig. 6 is proposed for SWR1 to hold either side of the linker DNA of the nucleosome. However, only 2 bp of the linker DNA at one side of the 133N2 nucleosome is available, which falls short of the stable binding of enzyme. Therefore, the model is not fully consistent with the idea of nucleosome flipping.

8. Tethering to the linker DNA through a DNA-binding element was found before in Chd1, which induces bidirectional DNA translocation of the nucleosome (doi: 10.1016/j.molcel.2017.08.018). The current model of Swr1 show a strong similarity to Chd1.

Minor points:

1. As shown in Fig.5, several DNA binding region of swc2 were discovered, the authors should show the local cryo-EM densities of these elements, so that the readers can judge the confidence of the model.
2. In Fig. 5a and 5c, the summarized cartoon of the cryo-EM structure are too small to read.
3. In Fig. 5c, the words 'SWR1-nucleosome in Configuration II' are covered partially by the figure.
4. In Fig. S12, the conserved sequence logo of contact #3 and # 4 did not point to the box regions.
5. The authors determined lifetimes of SWR1/nucleosome complexes using the strategy of SWR1 647N without any description of the experimental rationale (Fig. S2), which was discussed much later. A brief description of the experimental design will help the readers.
6. A dwell time analysis of the intermediate state was performed in Fig. 1d. But the physical/mechanistic meaning of the dwell time is not discussed. Does it depend on the ATP concentration of the exchange reaction?
7. Biphasic exponential kinetics was used to fit the data in S5. But the rationale to use biphasic exponential, rather than single exponential decay, is not discussed. The mechanistic implications of the fast and slow decays are not discussed either.

Referee #3 (Remarks to the Author):

The authors use extremely elegant single-molecule fluorescence experiments and sophisticated cryo-EM analyses to scrutinize SWR1-catalyzed histone exchange. They demonstrate that SWR1 alternates between engaging nucleosomes in two orientations without dissociating, with a strong preference for the WT H2A/H2B side in heterotypical nucleosomes containing one Htz1/H2B dimer. Additionally, single-molecule data provide strong evidence that SWR1 can catalyze the processive exchange of both H2A/H2B dimers. Analysis of less-populated states in a cryo-EM dataset revealed 2D classes that capture SWR1 in the process of flipping between different nucleosome sides, with the Swc2 subunit facilitating this process by interacting with linker DNA. The data presented in this manuscript are extremely timely, and of very high quality and outstanding interest to the chromatin and remodeler field. I am therefore happy to enthusiastically recommend publication of this manuscript, but only after the following points listed below have been addressed. Although I have suggested additional experiments to improve the manuscript, likely all of my points can be addressed textually.

Most importantly, the authors should reconcile their observations of processive histone exchange with their previous paper (PMID: 30309918), where (by analyzing histone exchange in bulk under catalytic conditions) they had provided strong evidence in favor of the distributive mechanism. In this earlier experiment, the authors had observed 69% of nucleosomes undergoing a single histone exchange, which seems to be in direct conflict with their new data.

One potential explanation could involve a facilitated diffusion/monkey-bar mechanism. Indeed, some indirect evidence supporting such a mechanism is that SWR1 has a residence time of tens of minutes in smFRET experiments, while it seems to be readily released by competitor DNA in bulk experiments. If k_{off} were as slow in bulk as in the smFRET experiments, one would not be able to stop the SWR1 reaction with competitor DNA in a timely fashion. The observation of the ability of SWR1 to flip between nucleosome sides by releasing its grip of the nucleosome while holding onto the linker DNA also agrees with a monkey-bar mechanism. In that case in single-molecule experiments, where nucleosomes are isolated from each other on the surface, SWR1 can stay bound to a single nucleosome for a long time, while in bulk it would hop between different nucleosomes. To probe this, the authors could consider adding unlabeled nucleosomes and/or competitor DNA in trans to see if that would affect the lifetime of SWR1 on nucleosomes in single-molecule experiments.

In the introduction the authors write "It is also unknown whether dimer exchange is a processive process, with both dimers exchanged in a nucleosome after a single SWR1 binding event, or is distributive with nucleosome release between dimer exchanges". Given the aforementioned previous publication from the same laboratories (PMID: 30309918), this statement should be corrected to accurately reflect existing data on SWR1 processivity.

A second important question concerns the kinetic proofreading. In my opinion, this term refers to a specific mechanism where energy expenditure is used to enhance the accuracy of substrate selection beyond the difference in affinity for correct and incorrect substrates. It is not clear from the text how the ability of SWR1 to flip between two orientations on a nucleosome with a strong preference for the WT dimer side enables a kinetic proofreading mechanism. The authors should clarify this.

The authors state that the preference of SWR1 to orient itself on heterotypical nucleosomes "...explain(s) the exquisite selectivity of the enzyme for replacing H2A with Htz and not the reverse". It makes perfect sense that the orientational bias plays an important role in such a selectivity mechanism. However, the observed preference seems far from absolute. It appears that SWR1 spends a significant fraction of time (on the order of 10%) in the undesired orientation, so the mechanism put forward by the authors would explain an approximately 10-fold preference for WT dimers over Htz1. Does that agree quantitatively with the experimental data? The authors describe the SWR1 preference as almost absolute. Could additional mechanisms be involved? The authors should consider either demonstrating the quantitative agreement or rephrasing this statement.

Minor points:

Some of the existing data suggest that Htz1 is enriched at the promoter-distal side of +1 nucleosomes (PMID: 25480300). Thus, SWR1 might have a preference for exchanging H2A/B dimers on the side of a nucleosome opposite the long linker. It would therefore be beneficial for readers to provide more context on the biological role of SWR1 and more specifically at promoters. There is contradictory data in the literature on whether linker DNA length affects the dimer exchange by SWR1 in a side-specific manner. The authors see no bias in dimer exchange in their experiments, while some previous studies have reported such a bias:

"The approximately equal areas under the two peaks further reveal equal propensity for exchange of each dimer in a yeast nucleosome, consistent with our previous single-molecule studies, although other studies using hybrid nucleosomes containing either *Xenopus* or *Drosophila* H3/H4 histones combined with yeast dimers show some asymmetry in dimer exchange."

This is an important question in the field, and it would be of great value if the authors could clarify these contradictory results. Since the authors imply that the use of H3/H4 histones from other species might be responsible for the apparent discrepancy, they should consider analyzing SWR1 histone exchange for such nucleosomes, too, if possible at all. More generally, the processive incorporation of Htz1 seems to be in conflict with it being enriched at the promoter-distal side of +1 nucleosomes. The authors should discuss this in the text.

"The ATPase domains of the INO80 subunit engage at superhelical location(SHL) 6" - a recent paper suggests that INO80 might be engaging at SHL2 (PMID: 35597239).

The authors should discuss the nature of nucleosomes which demonstrate no dynamics when bound to SWR1, and why their fraction seems to be very different for proximal and distal heterotypical nucleosomes. The time spent in an undesirable orientation seems to be substantially different for these two constructs. Also, the authors should add a distribution for WT nucleosomes to Fig. 4c, just like in Fig 4g.

The fact that the second exchange event during processive exchange takes much longer than the first is somewhat counterintuitive. The authors should discuss potential explanations. One of them could be that heterotypical nucleosomes have slower exchange rates. The authors could probe that by monitoring histone exchange on heterotypical nucleosomes in bulk or single-molecule experiments.

Author Rebuttals to Initial Comments:

Response to Referees' comments

Referee #1:

The paper from Girvan et al. is highly insightful and delves into the mechanism of H2A.Z exchange for H2A by the SWR1 chromatin remodeler using a variety of single molecule FRET based assays coupled with some cryoEM analysis. They provide evidence for SWR1 working processively such that after one dimer is replaced the enzyme remains bound and then subsequently exchanges the second dimer. The three-color FRET assay is particularly informative that allows dimer exchange to be tracked at the same time while following binding of SWR1 to nucleosomes.

The FRET assay with labeled nucleosome and SWR1 to detect flipping of the nucleosome when associated with SWR1 is well designed. However, I had a question as to how well the distal bound complex with SWR1 bound to the other side of the nucleosome away from the fluorescent donor is distinguished from background noise. It seems from the traces that the 0.1 FRET signal is so close to the noise level that it might not be that easy to distinguish the two from each other.

We agree that distinguishing 0.1 FRET from background can be challenging. However, we can readily resolve FRET ratios as low as 0.05. Below is a FRET histogram for a donor-only control with clearly distinguishable distributions. In the worst-case scenario, we could miss a small fraction of static low-FRET events, but importantly, this would not affect any of our kinetic analysis. To address this point, we have updated Extended Data Fig. 4 to include the donor-only control and a time trajectory that shows the FRET increase from 0 to 0.1 upon enzyme binding.

Response Figure 1. FRET histograms of donor-only control nucleosome (white) and flipping nucleosome (grey) from Figure 3c showing clearly distinguishable zero- and low-FRET states.

Although the simplest interpretation of these data is the orientation of the nucleosome is flipped could the authors also explain other potential interpretations of the data and why these would be less favored?

We have considered several alternative explanations, such as (i) DNA unwrapping, (ii) SWR1 binding and dissociating or (iii) SWR1 diffusing along the DNA. However, we have ruled out all of these options because (i) DNA unwrapping requires ATP binding [PMID: 30309918], (ii) flipping is still observed following SWR1 washout control and (iii) SWR1 diffusing along the DNA requires ATP binding [PMID 35876491]. To address this

point, we have clarified this in the main text (Page 9). See also the alternative labelling experiment suggested by Reviewer 2, point 2.

Could the authors explain why there is the discrepancy between their and others' reports as to whether the dimer proximal to the linker DNA is preferred to be exchanged or not. Is this the difference between yeast versus human/Xenopus hybrid nucleosomes and if so why is that?

This is an important point raised by all three reviewers and is addressed here to respond to all three. The first exchange reaction preference (or lack thereof) has been somewhat controversial for some time. Our initial smFRET studies [PMID: 30309918], showed a weak exchange preference (approximately 60:40 to 50:50 at low to high dimer:nucleosome ratios, respectively, Figure S14C therein) for the linker-distal (dye-proximal) dimer. Our new exchange data are consistent with those results (also approximately 55:45, Figure 1c herein). Furthermore, the Luk lab's room temperature data [PMID 31914392] are also consistent with a weak preference (approximately 55:45 for linker-distal, Figure 4D therein). Conversely, the Peterson/Loparo [PMID: 36396651] and Wu/Ha labs [PMID: 35263135] have reported a stronger preference for the linker-distal dimer (more frequent linker-distal exchanges, and 4-fold faster kinetics for linker-distal/dye-proximal exchange, respectively). We don't know the basis for this difference, but we speculate that the nature of the enzyme or histone source (i.e., yeast, frog, recombinant, etc) could contribute to these differences and/or the different nucleosome positioning sequences used by different labs.

Although we have failed to identify a specific reason, we reproducibly see the same weak bias in all our experiments. Importantly, all labs observe the same linker-distal preference, albeit to different bias extents (weak or stronger). In our hands, the weak bias we observe would not be significant. Moreover, the Luk lab data show that this preference dissipates at physiological temperatures (>23 °C). Lastly, although ChIP-exo data from yeast cells [PMID:25480300] show a marked preference for the linker-distal position at certain transcription start sites (linker-distal being denoted by the nucleosome free region), the same data also show a much weaker preference when determined globally (approximately 60:40, Figure 6D therein, see below), more similar to our own distribution. The corollary of the observed high linker-distal preference at certain sites suggests the opposite at other sites to maintain the overall 60:40 bias. Histone dimer exchange bias may play a role in the context of transcription, where the +1 nucleosome is flanked by chromatin on one side and a nucleosome-free region on the other. In the context of DNA repair, however, the nucleosome would likely be flanked by nucleosome-free regions on both sides, reducing possible histone dimer exchange bias. Nonetheless, we agree with the reviewer that this issue was not sufficiently discussed in the initial submission of the manuscript. Therefore, to address this concern, we have included these arguments in the discussion of the revised manuscript (Page 18).

[REDACTED]

In the heterotypic nucleosome FRET assays, could the linker DNA play a contributing role for SWR1 preferring to reside longer on the proximal side where the H2A/H2B resides? It would be important to reverse the nucleosome orientation such that the H2A/H2B dimer is on the long linker side of nucleosomes to confirm that linker DNA is not a contributing factor. I worry the linker DNA could be a factor based on how SWR1 interacts with the longer linker DNA.

The reviewer raises an interesting point. To address this, we have performed experiments with the linker DNA attached to the opposite end of the nucleosome. We still observe preferred binding to the face of the nucleosome containing the canonical H2A/H2B dimer (see Figure below), confirming our initial conclusions. We have added these data to the new Extended Data Figure 4k-n (copied below) and clarified this in the text (Page 11).

Response Figure 3. Panel from Extended Data Fig. 4. (k) cartoons of the 113N2 and swapped DNA 2N113 nucleosomes (l) Schematic of the flipping assay using the 2N113 heterotypic swapped DNA overhang nucleosome (m) FRET histogram of swapped linker nucleosome, showing SWR1 binding preference driven by H2A-H2B histone (low-FRET) over Htz1-H2B (mid-FRET). (n) Survival plot for the distal and proximal dwell times of a 2N113 heterotypic nucleosome, showing similar kinetics to 113N2 heterotypic nucleosome.

In the abstract the authors state that “CryoEM analysis reveals different populations of complexes showing how nucleosomes ‘flip’ between different conformation without release”. I think this is a bit of an overstatement because as discussed in the main text due to the dynamic flipping there are only to visualize well a small number of

conformations that are likely in the flipping state and are not sufficient to show how nucleosomes are flipped only that they have the potential to be.

The wording of the abstract has now been softened slightly to suggest views of the complex are seen that catch complexes in the act of flipping the nucleosomes rather than showing how this happens. Nonetheless, the cryo-EM analysis defines two states with 3D resolution, which we propose to be at each end of the flipping cycle and use these endpoints to extrapolate a mechanism to accommodate the flipping we observed in the single molecule experiments. The 2D classes within the Class II complexes (Extended Data Figs. 7 and 8) we observe are intermediates in this flipping process. However, since this is a dynamic process, there are multiple intermediate states in which the nucleosome core is released from the complex and are consequently in multiple different conformations in relation to the SWR1 complex. The consistent feature though is the observation of density connecting the nucleosome to the complex showing this DNA “leash” prevents complete disassociation of the nucleosome from the enzyme. Although we show just four particularly well defined 2D classes (and very obviously flipping complexes) in the main text and another four in the Extended Data Fig. 12, there are many more classes than this which show great diversity. The heterogeneity of this group is shown by the overall Class II averages in the Extended Data Figs. 7 and 8 which are actually the majority of complexes with nucleosome bound, again reflecting the dynamic nature of the complex.

Referee #2:

The exchange of the canonical H2A with the variant H2A.Z, Htz1 in yeast, alter the chromatin structure, and plays an important role in gene regulation. This manuscript by Girvan and colleagues explore the mechanism of Htz1-H2B dimer exchange by the yeast SWR1 complex using smFRET and cryoEM. The smFRET data showed that: 1. The reaction of double dimer exchange can be processive without the release of SWR1 from its substrate nucleosome; 2. SWR1 flips between the two faces of a nucleosome, which was regulated by the histone compositions. Additionally, they solved the cryo-EM structure of SWR1 bound to a nucleosome with the extended linker DNA bound by the Swc2 subunit.

The kinetics of the H2A.Z exchange reaction catalyzed by the SWR1 complex has been extensively studied, and the authors provide some new interesting findings. However, the current interpretations of the data at several places are not convincing.

Major points:

1. Using the 113N2 nucleosome as the substrate, the authors found the equal propensity for exchange of each dimer in a yeast nucleosome (Fig. 1c). This is at odd with the several earlier studies in vitro. The authors interpretate this discrepancy as the difference of the histones used. However, this interpretation is not convincing, as the failure to detect the linker DNA dependence of the exchange reaction may simply arise because of the other reagents used, including the DNA sequence, or other kinds of technical incompetence. Notably, the authors used recombinant SWR1 complex in this study, whereas the native complex was used by Poyton, Fan and Singh (ref. 18,19, and 20) in the previous studies. More importantly, the current finding is inconsistent with the asymmetric deposition of the Htz1 variant inside the cells.

Please see the response to Reviewer 1 (point 3, Page R2), who raised the same question.

2. The authors claimed that “we did not observe any distributive exchange”. But a distributive case was shown in Fig. S4G, and possibly another case in S4H. This raises a question how processive the enzyme can be, which is probably dependent on the experimental conditions, the salt concentration in particular.

The reviewer is correct. We do actually observe a small fraction (3 out of 25) of distributive trajectories and have corrected this mistake in the revised manuscript (p7). However, all processive enzymes will exhibit a fraction of distributive events depending on experimental conditions. The fact that we observe processive trajectories (15 out of 25) demonstrates that the enzyme can be processive and that is the crucial point as we need to explain how that can take place which this work answers.

3. In Fig.3, the authors claim that SWR1 flips between the two faces of a nucleosome. However, the altered FRET values can also be interpreted by the unwrapping or bending of the DNA. DNA bending and unwrapping are not the remote possibilities, considering the large conformational changes of the DNA bound by SWR1, as shown in Fig.5 and the previous study. I suggest the authors place the probes at more stable

positions, such as labelling the H2A-H2B dimer, and repeat the assay. Or the authors can provide evidences to eliminate these possibilities.

This point is addressed partially in the response to Reviewer 1, point 2. Nonetheless, as suggested by this reviewer, we repeated the experiment with the label (FRET donor) moved to H2A. The new data exhibit the same flipping dynamics (see Figure below), strongly supporting the nucleosome flipping interpretation of the original data. We have included these new data in Extended Data Fig. 4 (copied below) and incorporated this result in the revised manuscript (Page 9). We thank the reviewer for suggesting this important control.

Response Figure 3. Panel from Extended Data Fig. 4. (e) Nucleosome flipping experiment with modified donor position (on H2A^{K119C}). (f) FRET time trajectory showing flipping dynamics. (g) FRET histogram showing the two binding modes. Note that SWR1 still shows a weak preference for the linker-distal side (mid-FRET).

4. To indicate a different binding state in Fig. 3b, a low FRET value of 0.1 is used, which is, however, barely above the background noise. Quantification of the low/high FRET ratio is probably error prone.

Please see the response to Reviewer 1 (point 1), who raised the same question.

5. SWR1 is well known to selectively exchange the H2A-H2B for the Htz1-H2B. The findings in Fig. 4a-d provide little new insight into the selectivity, but is confirmatory to the previous studies. The stable binding to the hexasome shown in Fig. 4e-4f is very interesting. However, the authors offer little insights into this intermediate. As discussed above, a low FRET value of 0.1 for a prominent new state is less satisfactory. Placing the empty surface of the hexasome at the proximal site will provide much convincing evidence.

We agree with the reviewer that stable binding to a hexasome is very interesting. We have a follow-up paper describing in detail the structure of the hexasome-bound-SWR1 complex that is under review elsewhere. The structure shows that the hexasome does indeed bind as we suggest, with the “empty” site facing the enzyme. However, the follow-up experiment proposed here (placing the empty surface of the hexasome at the dye-proximal site) is not possible because the DNA would be unwrapped, making the FRET value too low to detect (near zero FRET).

6. A longer section of overhang DNA is evident that emanates from the the nucleosome. However, it is hard to tell by the current data shown in S11 A and B from which DNA

gyres the EM density come. The maps are noisy, and the connectivity does not seem to support a clear identity of the DNA gyres.

We have amended (what is now) Extended Data Fig. 10 to make the DNA gyres clearer. We also include two “rocking” movies (Supplementary Videos 1 and 2) that we hope show the DNA gyres even more clearly.

7. To explain the flipping behavior and the processivity observed in FRET experiments, a model in Fig. 6 is proposed for SWR1 to hold either side of the linker DNA of the nucleosome. However, only 2 bp of the linker DNA at one side of the 133N2 nucleosome is available, which falls short of the stable binding of enzyme. Therefore, the model is not fully consistent with the idea of nucleosome flipping.

We agree with the reviewer that this was not explained clearly enough in the original manuscript. Two possible (but very similar) models can facilitate nucleosome flipping: (i) SWR1 holds onto one of the DNA linkers and the nucleosome flips back and forth, or (ii) SWR1 sequentially switches between DNA linkers, flipping the nucleosome continuously in the same direction. Our single molecule data can only be consistent with model (i) as only one long DNA linker is present, but our data do not rule out model (ii) if both linkers are ~35 bp or longer (Fig. 5c and d). In the cell, the flipping mechanism may depend on the chromatin context. For example, in transcription with a single nucleosome-free region, model (i) may be favoured, whereas in repair, where two flanking nucleosome-free regions may be available, model (ii) may be favoured.

8. Tethering to the linker DNA through a DNA-binding element was found before in Chd1, which induces bidirectional DNA translocation of the nucleosome (doi: 10.1016/j.molcel.2017.08.018). The current model of Swr1 show a strong similarity to Chd1.

We agree with the reviewer that the two motors show some similarities although there are also a number of differences. Nonetheless, we have cited the Chd1 work accordingly (Page 19).

Minor points:

1. As shown in Fig.5, several DNA binding region of swc2 were discovered, the authors should show the local cryo-EM densities of these elements, so that the readers can judge the confidence of the model.

We have included representative density for the Swc2 contact regions (Extended Data Fig. 11b-e).

2. In Fig. 5a and 5c, the summarized cartoon of the cryo-EM structure are too small to read.

We have enlarged Fig. 5a and c to increase clarity.

3. In Fig. 5c, the words 'SWR1-nucleosome in Configuration II' are covered partially by the figure.

We have amended Fig. 5c accordingly.

4. In Fig. S12, the conserved sequence logo of contact #3 and #4 did not point to the box regions.

We have amended (what is now) Extended Data Fig.11a accordingly.

5. The authors determined lifetimes of SWR1/nucleosome complexes using the strategy of SWR1 647N without any description of the experimental rationale (Fig. S2), which was discussed much later. A brief description of the experimental design will help the readers.

For brevity in the main text, we described the experimental design in the caption of Extended Data Figure 2.

6. A dwell time analysis of the intermediate state was performed in Fig. 1d. But the physical/mechanistic meaning of the dwell time is not discussed. Does it depend on the ATP concentration of the exchange reaction?

The dwell time in Fig 1d relates to the time taken between exchange events. We are only able to monitor the step which is the actual exchange of the dimers. Other likely events include some sort of DNA unwinding to release the dimer which must take place during this interval and also (presumably) precedes the first exchange but our system does not (usually) allow us to determine that interval as most complexes are formed before measurements can begin.

The exchange reaction is ATP-dependent. No exchange is observed in the absence of ATP or in the presence of ATPγS. ATP is saturating in these conditions, increasing the concentration would be unlikely to change the dwell time. Lowering the ATP concentration would make the second exchange too slow to measure (relative to dye photobleaching).

7. Biphasic exponential kinetics was used to fit the data in S5. But the rationale to use biphasic exponential, rather than single exponential decay, is not discussed. The mechanistic implications of the fast and slow decays are not discussed either.

The revised manuscript (Page 30) now includes a discussion of the mechanistic implications of the double exponential kinetics.

Referee #3:

The authors use extremely elegant single-molecule fluorescence experiments and sophisticated cryo-EM analyses to scrutinize SWR1-catalyzed histone exchange. They demonstrate that SWR1 alternates between engaging nucleosomes in two orientations without dissociating, with a strong preference for the WT H2A/H2B side in heterotypical nucleosomes containing one Htz1/H2B dimer. Additionally, single-molecule data provide strong evidence that SWR1 can catalyze the processive exchange of both H2A/H2B dimers. Analysis of less-populated states in a cryo-EM dataset revealed 2D classes that capture SWR1 in the process of flipping between different nucleosome sides, with the Swc2 subunit facilitating this process by interacting with linker DNA. The data presented in this manuscript are extremely timely, and of very high quality and outstanding interest to the chromatin and remodeler field. I am therefore happy to enthusiastically recommend publication of this manuscript, but only after the following points listed below have been addressed. Although I have suggested additional experiments to improve the manuscript, likely all of my points can be addressed textually.

Most importantly, the authors should reconcile their observations of processive histone exchange with their previous paper (PMID: 30309918), where (by analyzing histone exchange in bulk under catalytic conditions) they had provided strong evidence in favor of the distributive mechanism. In this earlier experiment, the authors had observed 69% of nucleosomes undergoing a single histone exchange, which seems to be in direct conflict with their new data.

The reviewer is correct that there is an apparent discrepancy between our two manuscripts. The conclusion in our previous paper [PMID: 30309918] that SWR1 is distributive was reached under the assumption that, if both exchanges were fast and completely processive, the reaction would exhibit a high (up to 50%) proportion of double exchanges at 1:1 dimer-to-nucleosome ratio, which we did not observe (Fig. S14B therein). However, the new data reconciles this discrepancy because (i) the second exchange is much slower than the first one, and (ii) exchanges are heterogeneous, with a fraction (60%) of the observed exchanges being processive (Figure 2 and Extended Data Fig. 3). This would have been impossible to anticipate in the absence of the 3-color experiments presented here, thus emphasizing the need for these experiments. To address this important point, we have now included this discussion in the main text (Page 17).

One potential explanation could involve a facilitated diffusion/monkey-bar mechanism. Indeed, some indirect evidence supporting such a mechanism is that SWR1 has a residence time of tens of minutes in smFRET experiments, while it seems to be readily released by competitor DNA in bulk experiments. If koff were as slow in bulk as in the smFRET experiments, one would not be able to stop the SWR1 reaction with competitor DNA in a timely fashion. The observation of the ability of SWR1 to flip between nucleosome sides by releasing its grip of the nucleosome while holding onto the linker DNA also agrees with a monkey-bar mechanism. In that case in single-molecule experiments, where nucleosomes are isolated from each other on the surface, SWR1 can stay bound to a single nucleosome for a long time, while in bulk it would hop

between different nucleosomes. To probe this, the authors could consider adding unlabeled nucleosomes and/or competitor DNA in trans to see if that would affect the lifetime of SWR1 on nucleosomes in single-molecule experiments.

This is an interesting suggestion. To address this, we carried out the proposed experiment in the presence of unlabelled competitor nucleosomes and found no difference in the lifetime of the SWR1-nucleosome complex (18 min vs 19 min in Extended Data Figure 2), suggesting this is not the case.

Response Figure 5. SWR1-nucleosome complex lifetime in the presence of unlabelled competitor nucleosomes.

In the introduction the authors write “It is also unknown whether dimer exchange is a processive process, with both dimers exchanged in a nucleosome after a single SWR1 binding event, or is distributive with nucleosome release between dimer exchanges”. Given the aforementioned previous publication from the same laboratories (PMID: 30309918), this statement should be corrected to accurately reflect existing data on SWR1 processivity.

As discussed above, we have now addressed this in the revised manuscript (Pages 4 and 17).

A second important question concerns the kinetic proofreading. In my opinion, this term refers to a specific mechanism where energy expenditure is used to enhance the accuracy of substrate selection beyond the difference in affinity for correct and incorrect substrates. It is not clear from the text how the ability of SWR1 to flip between two orientations on a nucleosome with a strong preference for the WT dimer side enables a kinetic proofreading mechanism. The authors should clarify this.

According to Hopfield and Ninio, proofreading mechanisms increase specificity in biochemical reactions by allowing for the dissociation of (incorrect) intermediate complexes. Hopfield proposed a mechanism for biosynthetic pathways that required a rapid, energy expenditure step (e.g. nucleotide hydrolysis) to drive the proofreading.

However, Ninio's more general model refers instead to a "sticking time". The basic requirement is that steps after the proofreading step should be slow compared to that step, and the proofreading effect is the ratio between the exit rates from each state. For SWR1, dimer exchange likely requires multiple ATPase steps involving DNA translocation/unwinding before the dimer can be released. The dimer exchange certainly takes considerably longer than flipping, so meets this criterion. The fast dissociation and flipping of SWR1 from the exchanged side to the canonical side has the same proofreading effect as dissociating from an incorrect substrate. We agree with the reviewer that this was not made clear in the original manuscript, and we have modified this in the revised manuscript (Pages 11-12) to make this clearer.

The authors state that the preference of SWR1 to orient itself on heterotypical nucleosomes "...explain(s) the exquisite selectivity of the enzyme for replacing H2A with Htz and not the reverse". It makes perfect sense that the orientational bias plays an important role in such a selectivity mechanism. However, the observed preference seems far from absolute. It appears that SWR1 spends a significant fraction of time (on the order of 10%) in the undesired orientation, so the mechanism put forward by the authors would explain an approximately 10-fold preference for WT dimers over Htz1. Does that agree quantitatively with the experimental data? The authors describe the SWR1 preference as almost absolute. Could additional mechanisms be involved? The authors should consider either demonstrating the quantitative agreement or rephrasing this statement.

We agree with the reviewer that although orientational bias plays an important role in selectivity, this may not be absolute. Additional mechanisms, such as recognition of the alpha-2 helix in H2A (PMID: 26116819), may also be involved. We have changed the word "explaining" to "contributing to" and discussed the whole issue further in the revised manuscript as a part of addressing the previous point (Page 12).

Minor points:

Some of the existing data suggest that Htz1 is enriched at the promoter-distal side of +1 nucleosomes (PMID: 25480300). Thus, SWR1 might have a preference for exchanging H2A/B dimers on the side of a nucleosome opposite the long linker. It would therefore be beneficial for readers to provide more context on the biological role of SWR1 and more specifically at promoters. There is contradictory data in the literature on whether linker DNA length affects the dimer exchange by SWR1 in a side-specific manner. The authors see no bias in dimer exchange in their experiments, while some previous studies have reported such a bias:

"The approximately equal areas under the two peaks further reveal equal propensity for exchange of each dimer in a yeast nucleosome, consistent with our previous single-molecule studies, although other studies using hybrid nucleosomes containing either Xenopus or Drosophila H3/H4 histones combined with yeast dimers show some asymmetry in dimer exchange."

This is an important question in the field, and it would be of great value if the authors could clarify these contradictory results. Since the authors imply that the use of H3/H4 histones from other species might be responsible for the apparent discrepancy, they

should consider analyzing SWR1 histone exchange for such nucleosomes, too, if possible at all. More generally, the processive incorporation of Htz1 seems to be in conflict with it being enriched at the promoter-distal side of +1 nucleosomes. The authors should discuss this in the text.

Please see the response to Reviewer 1 (point 3), who raised the same question.

“The ATPase domains of the INO80 subunit engage at superhelical location(SHL) 6” - a recent paper suggests that INO80 might be engaging at SHL2 (PMID: 35597239).

INO80 binding to SHL2 has only been observed in the context of hexasome binding and sliding. However, we have a manuscript under consideration elsewhere showing that INO80 can slide crosslinked nucleosomes, demonstrating that transient hexasome formation is not a required intermediate for nucleosome sliding, and its mechanistic relevance is unclear. This result implies that INO80 must engage with SHL6 and not SHL2 in a remarkably different mechanism than SWR1, which does require hexasome intermediate formation.

The authors should discuss the nature of nucleosomes which demonstrate no dynamics when bound to SWR1, and why their fraction seems to be very different for proximal and distal heterotypic nucleosomes. The time spent in an undesirable orientation seems to be substantially different for these two constructs. Also, the authors should add a distribution for WT nucleosomes to Fig. 4c, just like in Fig 4g.

We expect to see a small fraction of static traces that photobleach or dissociate before flipping. To illustrate this, we have added rastergrams to Extended Data Figure 5 that summarize all the single-molecule trajectories. Heterotypic nucleosomes are synthesized from hexasomes, and therefore, there can be a small fraction (~5-10%) of hexasome contamination in the prep, which would skew the fraction of static traces. Nonetheless, neither of these two scenarios would significantly affect the dwell time analysis. We have clarified this in the Methods. As requested, we have also added the dotted line to Fig. 4C.

The fact that the second exchange event during processive exchange takes much longer than the first is somewhat counterintuitive. The authors should discuss potential explanations. One of them could be that heterotypic nucleosomes have slower exchange rates. The authors could probe that by monitoring histone exchange on heterotypic nucleosomes in bulk or single-molecule experiments.

We were equally surprised by this counterintuitive result. As suggested, we have now performed single-molecule exchange experiments on heterotypic nucleosomes and found that the exchange time is ~100s, still significantly slower than the first exchange of canonical nucleosomes (~30s). This is in qualitative agreement with bulk experiments from the Peterson lab, also showing that the second exchange is slower [PMID 30970243]. Thus, the reviewer’s suggestion is correct that heterotypic nucleosomes have slower exchange rates and we thank them for suggesting the

experiment. We have included these data in the revised manuscript (Page 17 and Extended Data Figure 3).

Reviewer Reports on the First Revision:

Referees' comments:

Referee #1 (Remarks to the Author):

The authors have addressed well many of my original questions. The cryo-EM structures and how they relate to flipping of the SWR1 complex on nucleosomes is, however, still a concern. The two configurations they focus on show that SWR binds to the linker DNA and nucleosomal DNA with two alternative patterns without changing the SWR1 orientation relative to the nucleosome (Fig. 5a-d). Although the authors interpret these structures in line with the flipping of SWR relative to the faces of the nucleosome, I don't think that is completely the case. If SWR1 binding to DNA and switching its contacts on DNA indeed facilitated SWR1 flipping its orientation on nucleosomes, then why don't we see the orientation of the SWR1 relative to nucleosomes change in these two different conformations of the complex? In Fig. 6c, they propose that these two conformations represent SWR1 flipping its orientation relative to the nucleosome with the intermediate step being where the nucleosomes are completely released which then allows them to reposition. The problem with their model is that in the configurations they observed by cryo-EM SWR1 don't change its orientation on the surface of the nucleosome. I think the extrapolation and interpretation of the cryo-EM data raises important concerns and are inconsistent.

I appreciate the response provided by the authors explaining how other potential explanations of how the smFRET data could occur without SWR flipping its orientation. They point to DNA unwrapping or SWR1 diffusing along DNA not being likely because they both require SWR1 to bind ATP; however, the two cryo-EM structures mentioned above seems to clearly show that DNA near the edge of nucleosomes can in fact be bound in two alternative configurations. My question therefore is could these alternative configurations potentially explain the smFRET data and be an alternative explanation instead of SWR flipping on nucleosomes?

The authors effectively addressed the question as to the differences that exist in the literature on whether the distal or proximal dimer face is the preferred site for dimer exchange and that the question cannot be fully resolved at this time. The new experiment where the linker DNA is switched to the other side addressed well the question raised by several of the reviewers including myself.

Referee #2 (Remarks to the Author):

The authors answered my questions fully. I have no more question,

Referee #3 (Remarks to the Author):

The authors did an excellent job addressing all the points raised by the reviewers. The revised manuscript is even better as the result, and I therefore enthusiastically recommend the publication of this work.

Author Rebuttals to First Revision:

Referee 1:

The authors have addressed well many of my original questions. The cryo-EM structures and how they relate to flipping of the SWR1 complex on nucleosomes is, however, still a concern. The two configurations they focus on show that SWR binds to the linker DNA and nucleosomal DNA with two alternative patterns without changing the SWR1 orientation relative to the nucleosome (Fig. 5a-d). Although the authors interpret these structures in line with the flipping of SWR relative to the faces of the nucleosome, I don't think that is completely the case. If SWR1 binding to DNA and switching its contacts on DNA indeed facilitated SWR1 flipping its orientation on nucleosomes, then why don't we see the orientation of the SWR1 relative to nucleosomes change in these two different conformations of the complex? In Fig. 6c, they propose that these two conformations represent SWR1 flipping its orientation relative to the nucleosome with the intermediate step being where the nucleosomes are completely released which then allows them to reposition. The problem with their model is that in the configurations they observed by cryo-EM SWR1 don't change its orientation on the surface of the nucleosome. I think the extrapolation and interpretation of the cryo-EM data raises important concerns and are inconsistent.

I appreciate the response provided by the authors explaining how other potential explanations of how the smFRET data could occur without SWR flipping its orientation. They point to DNA unwrapping or SWR1 diffusing along DNA not being likely because they both require SWR1 to bind ATP; however, the two cryo-EM structures mentioned above seems to clearly show that DNA near the edge of nucleosomes can in fact be bound in two alternative configurations. My question therefore is could these alternative configurations potentially explain the smFRET data and be an alternative explanation instead of SWR flipping on nucleosomes?

The authors effectively addressed the question as to the differences that exist in the literature on whether the distal or proximal dimer face is the preferred site for dimer exchange and that the question cannot be fully resolved at this time. The new experiment where the linker DNA is switched to the other side addressed well the question raised by several of the reviewers including myself.

We are pleased that the reviewer recognises that we have addressed their original questions well, that they find the new experiments with the swapped linker convincing, and that we have addressed effectively the question of the exchange preference.

The reviewer raises a new concern regarding the two configurations (I and II) that we observe in cryoEM. The reviewer's suggestion is that the DNA overhang might be moving between the two configurations while the nucleosome itself remains static, rather than our interpretation that the nucleosomes are flipping and these two configurations are intermediates on that pathway. This might be a reasonable interpretation in the absence of the SM data, but we rule out this interpretation by showing that moving the label from the DNA to one of the two H2A subunits (thereby making the nucleosome asymmetric) also shows the same nucleosome flipping dynamics (Extended Data Fig. 5a-d). The reviewer's interpretation of DNA overhang flipping between the two configurations while the nucleosome remains static is not consistent with those observations. The problem is that cryoEM data only gives a static picture of states, and two states in which the two faces of the nucleosome are symmetrical, but inverted, cannot be distinguished from one another. That is precisely why it is important to combine the (dynamic) single molecule data with the (static) high resolution structural data. The synergy of these two approaches allows determination of a mechanism that would not be evident from either technique alone and rules out alternative interpretations. From the structural perspective, the observation that the binding of the nucleosome body is identical in the two states is exactly as expected since the nucleosome is symmetrical with respect to the histone contacts on the faces. The SM data provide evidence to demonstrate the flipping, while the structural data traps states on that pathway to allow us to interpolate between them and propose a mechanism. The movie we provided with the revised manuscript shows how flipping could take place and shows that rather than the DNA overhang flipping, one of these remains relatively static and associated with Swc5. Indeed the "red" overhang in Configuration II becomes the "blue" overhang in Configuration I without relocating but only as a consequence of flipping.

Nonetheless, to clarify this point, we have added the following explanation on Page 14:

"Due to the symmetry of the histone octamer, we cannot distinguish whether the nucleosome orientation relative to SWR1 switches between configurations I and II. An alternative explanation could be that SWR1

remains bound to the same face of the nucleosome in both configurations, and only the DNA overhang interacting with Swc2 is swapped. However, the smFRET experiments in which the donor is located on H2A still exhibit the nucleosome flipping dynamics (Extended Data Fig. 4e-g), thereby ruling out this possibility. Consequently, we interpret the two major states we observe by cryoEM to represent intermediates on the flipping pathway.”

Reviewer Reports on the Second Revision:

Referees' comments:

Referee #1 (Remarks to the Author):

I appreciate and accept the authors' explanation of the cryoEM data and how the single molecule data helps address some of the ambiguity of the structural data